# Generalized Data Weighting via Class-level Gradient Manipulation

**Can Chen[1]\*, Shuhao Zheng[1]\*, Xi Chen[1], Erqun Dong[1], Xue Liu[1], Hao Liu[2], Dejing Dou[3]**

[1]McGill University, [2]The Hong Kong University of Science and Technology, [3]Baidu Research
{can.chen, shuhao.zheng, erqun.dong}@mail.mcgill.ca
xi.chen11@mcgill.ca, xueliu@cs.mcgill.ca, liuh@ust.hk, doudejing@baidu.com

## Abstract

Label noise and class imbalance are two major issues coexisting in real-world datasets. To alleviate the two issues, state-of-the-art methods reweight each instance by leveraging a small amount of clean and unbiased data. Yet, these methods overlook class-level information within each instance, which can be further utilized to improve performance. To this end, in this paper, we propose **G**eneralized **D**ata **W**eighting (**GDW**) to simultaneously mitigate label noise and class imbalance by manipulating gradients at the class level. To be specific, GDW unrolls the loss gradient to class-level gradients by the chain rule and reweights the flow of each gradient separately. In this way, GDW achieves remarkable performance improvement on both issues. Aside from the performance gain, GDW efficiently obtains class-level weights without introducing any extra computational cost compared with instance weighting methods. Specifically, GDW performs a gradient descent step on class-level weights, which only relies on intermediate gradients. Extensive experiments in various settings verify the effectiveness of GDW. For example, GDW outperforms state-of-the-art methods by $2.56\%$ under the $60\%$ uniform noise setting in CIFAR10. Our code is available at https://github.com/GGchen1997/GDW-NIPS2021.

## 1  Introduction

Real-world classification datasets often suffer from two issues, i.e., label noise [1] and class imbalance [2]. On the one hand, label noise often results from the limitation of data generation, e.g., sensor errors [3] and mislabeling from crowdsourcing workers [4]. Label noise misleads the training process of DNNs and degrades the model performance in various aspects [5, 6, 7]. On the other hand, imbalanced datasets are either naturally long-tailed [8, 9] or biased from the real-world distribution due to imperfect data collection [10, 11]. Training with imbalanced datasets usually results in poor classification performance on weakly represented classes [12, 13, 14]. Even worse, these two issues often coexist in real-world datasets [15].

To prevent the model from memorizing noisy information, many important works have been proposed, including label smoothing [16], noise adaptation [17], importance weighting [18], GLC [19], and Co-teach [20]. Meanwhile, [12, 13, 14, 21] propose effective methods to tackle class imbalance. However, these methods inevitably introduce hyper-parameters (e.g., the weighting factor in [13] and the focusing parameter in [21]), compounding real-world deployment.

Inspired by recent advances in meta-learning, some works [22, 23, 24, 25] propose to solve both issues by leveraging a clean and unbiased meta set. These methods treat instance weights as hyper-

---

*Equal contribution; Names listed in alphabetical order.

35th Conference on Neural Information Processing Systems (NeurIPS 2021).

parameters and dynamically update these weights to circumvent hyper-parameter tuning. Specifically, MWNet [23] adopts an MLP with the instance loss as input and the instance weight as output. Due to the MLP, MWNet has better scalability on large datasets compared with INSW [24] which assigns each instance with a learnable weight. Although these methods can handle label noise and class imbalance to some extent, they cannot fully utilize class-level information within each instance, resulting in the potential loss of useful information. For example, in a three-class classification task, every instance has three logits. As shown in Figure 1, every logit corresponds to a class-level gradient flow which stems from the loss function and back-propagates. These gradient flows represent three kinds of information: "not cat", "dog", and "not bird". Instance weighting methods [23, 22] alleviate label noise by downweighting all the gradient flows of the instance, which discards three kinds of information simultaneously. Yet, downweighting the "not bird" gradient flow is a waste of information. Similarly, in class imbalance scenarios, different gradient flows represent different class-level information.

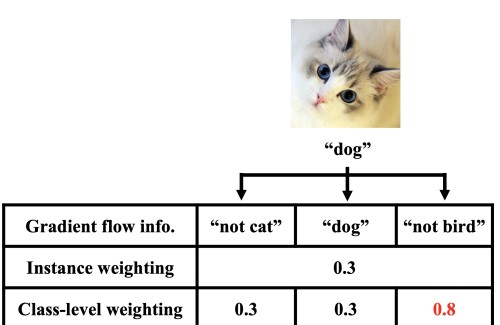

| Gradient flow info. | "not cat" | "dog" | "not bird" |
|---|---|---|---|
| Instance weighting | 0.3 | | |
| Class-level weighting | 0.3 | 0.3 | 0.8 |

Figure 1: Motivation for class-level weighting. For a noisy instance (e.g. cat mislabeled as "dog"), all gradient flows are downweighted by instance weighting. Although the gradient flows for "dog" and "not cat" contain harmful information, the gradient flow for "not bird" is still valuable for training, which should not be downweighted.

Therefore, it is necessary to reweight instances at the class level for better information usage.

To this end, we propose Generalized Data Weighting (**GDW**) to tackle label noise and class imbalance by class-level gradient manipulation. Firstly, we introduce class-level weights to represent the importance of different gradient flows and manipulate the gradient flows with these class-level weights. Secondly, we impose a zero-mean constraint on class-level weights for stable training. Thirdly, to efficiently obtain class-level weights, we develop a two-stage weight generation scheme embedded in bi-level optimization. As a side note, the instance weighting methods [22, 23, 24, 25] can be considered special cases of GDW when class-level weights within any instance are the same. In this way, GDW achieves impressive performance improvement in various settings.

To sum up, our contribution is two-fold:

1. For better information utilization, we propose GDW, a generalized data weighting method, which better handles label noise and class imbalance. To the best of our knowledge, we are the first to propose single-label class-level weighting on gradient flows.

2. To obtain class-level weights efficiently, we design a two-stage scheme embedded in a bi-level optimization framework, which does not introduce any extra computational cost. To be specific, during the back-propagation we store intermediate gradients, with which we update class-level weights via a gradient descent step.

## 2 Related Works

### 2.1 Non-Meta-Learning Methods for Label Noise

Label noise is a common problem in classification tasks [6, 7, 5]. To avoid overfitting to label noise, [16] propose label smoothing to regularize the model. [17, 26] form different models to indicate the relation between noisy instances and clean instances. [18] estimate an importance weight for each instance to represent its value to the model. [20] train two models simultaneously and let them teach each other in every mini-batch. However, without a clean dataset, these methods cannot handle severe noise [22]. [19] correct the prediction of the model by estimating the label corruption matrix via a clean validation set, but this matrix is the same across all instances. Instead, our method generates dynamic class-level weights for every instance to improve training.

### 2.2 Non-Meta-Learning Methods for Class Imbalance

Many important works have been proposed to handle class imbalance [27, 28, 29, 12, 30, 13, 31, 21, 14]. [28, 29] propose to over-sample the minority class and under-sample the majority class. [27, 30]

Table 1: Related works comparison. "Noise" and "Imbalance" denote whether the method can solve label noise and class imbalance. "Class-level" denotes whether the method utilizes class-level information in each instance, and "Scalability" denotes whether the method can scale to large datasets.

| | Focal [21] | Balanced [13] | Co-teaching [20] | GLC [19] | L2RW [22] | INSW [24] | MWNet [23] | Soft-label [42] | Gen-label [40] | **GDW** |
|---|---|---|---|---|---|---|---|---|---|---|
| Noise | ✗ | ✗ | ✓ | ✓ | ✓ | ✓ | ✓ | ✓ | ✓ | ✓ |
| Imbalance | ✓ | ✓ | ✗ | ✗ | ✓ | ✓ | ✓ | ✗ | ✗ | ✓ |
| Class-level | ✗ | ✗ | ✗ | ✗ | ✗ | ✗ | ✗ | ✓ | ✓ | ✓ |
| Scalability | ✓ | ✓ | ✓ | ✓ | ✓ | ✗ | ✓ | ✗ | ✗ | ✓ |

learn a class-dependent cost matrix to obtain robust representations for both majority and minority classes. [12, 13, 21, 14] design a reweighting scheme to rebalance the loss for each class. These methods are quite effective, whereas they need to manually choose loss functions or hyper-parameters. [32, 33] manipulate the feature space to handle class imbalance while introducing extra model parameters. [31] decouple representation learning and classifier learning on long-tailed datasets, but with extra hyper-parameter tuning. In contrast, meta-learning methods view instance weights as hyper-parameters and dynamically update them via a meta set to avoid hyper-parameter tuning.

## 2.3 Meta-Learning Methods

With recent development in meta-learning [34, 35, 36], many important methods have been proposed to handle label noise and class imbalance via a meta set [37, 38, 22, 39, 23, 24, 25, 40]. [38] propose MentorNet to provide a data-driven curriculum for the base network to focus on correct instances. To distill effective supervision, [41] estimate pseudo labels for noisy instances with a meta set. To provide dynamic regularization, [42, 40] treat labels as learnable parameters and adapt them to the model's state. Although these methods can tackle label noise, they introduce huge amounts of learnable parameters and thus cannot scale to large datasets. To alleviate class imbalance, [37] describe a method to learn from long-tailed datasets. Specifically, [37] propose to encode meta-knowledge into a meta-network and model the tail classes by transfer learning.

Furthermore, many meta-learning methods propose to mitigate the two issues by reweighting every instance [22, 43, 23, 24, 25]. [43] equip each instance and each class with a learnable parameter to govern their importance. By leveraging a meta set, [22, 23, 24, 25] learn instance weights and model parameters via bi-level optimization to tackle label noise and class imbalance. [22] assign weights to training instances only based on their gradient directions. Furthermore, [24] combine reinforce learning and meta-learning, and treats instance weights as rewards for optimization. However, since each instance is directly assigned with a learnable weight, INSW can not scale to large datasets. Meanwhile, [23, 25] adopt a weighting network to output weights for instances and use bi-level optimization to jointly update the weighting network parameters and model parameters. Although these methods handle label noise and class imbalance by reweighting instances, a scalar weight for every instance cannot capture class-level information, as shown in Figure 1. Therefore, we introduce class-level weights for different gradient flows and adjust them to better utilize class-level information.

We show the differences between GDW and other related methods in Table 1.

## 3 Method

### 3.1 Notations

In most classification tasks, there is a training set $D_{train} = \{(x_i, y_i)\}_{i=1}^{N}$ and we assume there is also a clean unbiased meta set $D_{meta} = \{(x_i^v, y_i^v)\}_{i=1}^{M}$. We aim to alleviate label noise and class imbalance in $D_{train}$ with $D_{meta}$. The model parameters are denoted as $\boldsymbol{\theta}$, and the number of classes is denoted as $C$.

### 3.2 Class-level Weighting by Gradient Manipulation

To utilize class-level information, we learn a class-level weight for every gradient flow instead of a scalar weight for all $C$ gradient flows in [23]. Denote $\mathcal{L}$ as the loss of any instance. Applying the

chain rule, we unroll the gradient of $\mathcal{L}$ w.r.t. $\boldsymbol{\theta}$ as

$$\nabla_{\boldsymbol{\theta}}\mathcal{L} = \frac{\partial\mathcal{L}}{\partial\boldsymbol{\theta}} = \frac{\partial\mathcal{L}}{\partial \mathbf{l}}\frac{\partial \mathbf{l}}{\partial\boldsymbol{\theta}} \doteq \mathbf{D}_1\mathbf{D}_2, \tag{1}$$

where $\mathbf{l} \in \mathbb{R}^C$ represents the predicted logit vector of the instance. We introduce class-level weights $\boldsymbol{\omega} \in \mathbb{R}^C$ and denote the $j^{th}$ component of $\boldsymbol{\omega}$ as $\boldsymbol{\omega}_j$. To indicate the importance of every gradient flow, we perform an element-wise product $f_{\boldsymbol{\omega}}(\cdot)$ on $\mathbf{D}_1$ with $\boldsymbol{\omega}$. After this manipulation, the gradient becomes

$$f_{\boldsymbol{\omega}}\left(\nabla_{\boldsymbol{\theta}}\mathcal{L}\right) \doteq \left(\boldsymbol{\omega} \otimes \frac{\partial\mathcal{L}}{\partial \mathbf{l}}\right)\frac{\partial \mathbf{l}}{\partial\boldsymbol{\theta}} = \left(\boldsymbol{\omega} \otimes \mathbf{D}_1\right)\mathbf{D}_2 \doteq \mathbf{D}_1'\mathbf{D}_2, \tag{2}$$

where $\otimes$ denotes the element-wise product of two vectors. Note that $\boldsymbol{\omega}_j$ represents the importance of the $j^{th}$ gradient flow. Obviously, instance weighting is a special case of GDW when elements of $\boldsymbol{\omega}$ are the same. Most classification tasks [44, 45, 46] adopt the *Softmax-CrossEntropy* loss. In this case, we have $\mathbf{D}_1 = \mathbf{p} - \mathbf{y}$, where $\mathbf{p} \in \mathbb{R}^C$ denotes the probability vector output by *softmax* and $\mathbf{y} \in \mathbb{R}^C$ denotes the one-hot label of the instance (see Appendix A for details).

As shown in Figure 1, for a noisy instance (e.g., cat mislabeled as "dog"), instance weighting methods assign a low scalar weight to all gradient flows of the instance. Instead, GDW assigns class-level weights to different gradient flows by leveraging the meta set. In other words, GDW tries to downweight the gradient flows for "dog" and "not cat", and upweight the gradient flow for "not bird". Similarly, in imbalance settings, different gradient flows have different class-level information. Thus GDW can also better handle class imbalance by adjusting the importance of different gradient flows.

### 3.3 Zero-mean Constraint on Class-level Weights

To retain the *Softmax-CrossEntropy* loss structure, i.e. the $\boldsymbol{p} - \boldsymbol{y}$ form, after the manipulation, we impose a zero-mean constraint on $\mathbf{D}_1'$. That is, we analyze the $j^{th}$ element of $\mathbf{D}_1'$ (see Appendix B.1 for details):

$$\boldsymbol{\omega}_j(\boldsymbol{p}_j - \mathbf{y}_j) = \boldsymbol{\omega}_t\left(\boldsymbol{p}_j' - \mathbf{y}_j\right) + \left(\sum_k \boldsymbol{\omega}_k\boldsymbol{p}_k - \boldsymbol{\omega}_t\right)\boldsymbol{p}_j', \tag{3}$$

where $\boldsymbol{p}_j' \doteq \frac{\boldsymbol{\omega}_j\boldsymbol{p}_j}{\sum_k \boldsymbol{\omega}_k\boldsymbol{p}_k}$ is the weighted probability, and $\boldsymbol{\omega}_t$ denotes the class-level weight at the target (label) position. We observe that the first term in Eq. (3) satisfies the structure of the gradient of the *Softmax-CrossEntropy* loss, and thus propose to eliminate the second term which messes the structure. Specifically, we let

$$\sum_k \boldsymbol{\omega}_k\boldsymbol{p}_k - \boldsymbol{\omega}_t = 0 \Rightarrow \boldsymbol{\omega}_t = \frac{\sum_{j \neq t}\boldsymbol{\omega}_j\boldsymbol{p}_j}{1 - \boldsymbol{p}_t}, \tag{4}$$

where $\boldsymbol{p}_t$ is the probability of the target class. Note that $\sum_j \boldsymbol{\omega}_j\mathbf{y}_j = \boldsymbol{\omega}_t$, and thus we have

$$\sum_j \boldsymbol{\omega}_j(\boldsymbol{p}_j - \mathbf{y}_j) = 0. \tag{5}$$

This restricts the mean of $\mathbf{D}_1'$ to be zero. Therefore, we name this constraint as the **zero-mean constraint**. With this, we have

$$\mathbf{D}_1' = \boldsymbol{\omega}_t\left(\boldsymbol{p}' - \mathbf{y}\right). \tag{6}$$

Eq. (6) indicates that $\boldsymbol{\omega}$ adjust the gradients in two levels, i.e., instance level and class level. Namely, the scalar $\boldsymbol{\omega}_t$ acts as the instance-level weight in previous instance weighting methods [22, 23, 24, 25], and the $\boldsymbol{\omega}_j$'s are the class-level weights manipulating gradient flows by adjusting the probability from $\mathbf{p}$ to $\mathbf{p}'$.

### 3.4 Efficient Two-stage Weight Generation Embedded in Bi-level Optimization

In this subsection, we first illustrate the three-step bi-level optimization framework in [23]. Furthermore, we embed a two-stage scheme in the bi-level optimization framework to efficiently obtain class-level weights, with which we manipulate gradient flows and optimize model parameters.

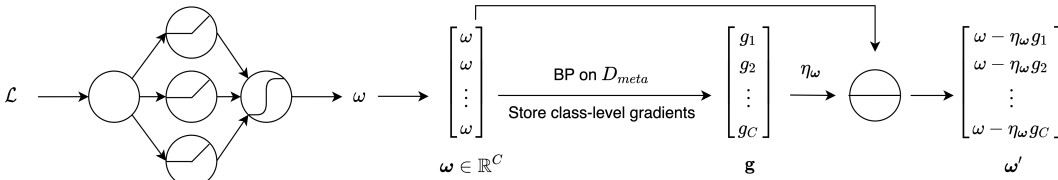

Figure 2: Two-stage Weight Generation. "BP" denotes the back-propagation in **Step 2** of the bi-level optimization framework. $\mathbf{g}$ denotes the intermediate gradients w.r.t. $\boldsymbol{\omega}$. $\ominus$ denotes the minus operator. Note that $\boldsymbol{\omega}$ is the first-stage (instance-level) weight and $\boldsymbol{\omega}'$ is the second-stage (class-level) weight.

**Three-step Bi-level Optimization.** Generally, the goal of classification tasks is to obtain the optimal model parameters $\boldsymbol{\theta}^*$ by minimizing the average loss on $D_{train}$, denoted as $\frac{1}{N}\sum_{i=1}^{N} l_{train}(x_i, y_i; \boldsymbol{\theta})$. As an instance weighting method, [23] adopt a three-layer MLP parameterized by $\phi$ as the weighting network and take the loss of the $i^{th}$ instance as input and output a scalar weight $\omega_i$. Then $\boldsymbol{\theta}^*$ is optimized by minimizing the instance-level weighted training loss:

$$\boldsymbol{\theta}^*(\phi) = \arg\min_{\boldsymbol{\theta}} \frac{1}{N} \sum_{i=1}^{N} \omega_i(\phi) l_{train}(x_i, y_i; \boldsymbol{\theta}). \tag{7}$$

To obtain the optimal $\omega_i$, they propose to use a meta set as meta-knowledge and minimize the meta-loss to obtain $\phi^*$:

$$\phi^* = \arg\min_{\phi} \frac{1}{M} \sum_{i=1}^{M} l_{val}(x_i^v, y_i^v; \boldsymbol{\theta}^*(\phi)). \tag{8}$$

Since the optimization for $\boldsymbol{\theta}^*(\phi)$ and $\phi^*$ is nested, they adopt an online strategy to update $\boldsymbol{\theta}$ and $\phi$ with a three-step optimization loop for efficiency. Denote the two sets of parameters at the $\tau^{th}$ loop as $\boldsymbol{\theta}_\tau$ and $\phi_\tau$ respectively, and then the three-step loop is formulated as:

**Step 1** Update $\boldsymbol{\theta}_{\tau-1}$ to $\hat{\boldsymbol{\theta}}_\tau(\phi)$ via an SGD step on a mini-batch training set by Eq. (7).

**Step 2** With $\hat{\boldsymbol{\theta}}_\tau(\phi)$, update $\phi_{\tau-1}$ to $\phi_\tau$ via an SGD step on a mini-batch meta set by Eq. (8).

**Step 3** With $\phi_\tau$, update $\boldsymbol{\theta}_{\tau-1}$ to $\boldsymbol{\theta}_\tau$ via an SGD step on the same mini-batch training set by Eq. (7).

Instance weights in **Step 3** are better than those in **Step 1**, and thus are used to update $\boldsymbol{\theta}_{\tau-1}$.

**Two-stage Weight Generation.** To guarantee scalability, we apply the same weighting network in [23] to obtain weights. To efficiently train $\phi$ and $\boldsymbol{\theta}$, we also adopt the three-step bi-level optimization framework. Moreover, we propose an efficient two-stage scheme embedded in **Step 1-3** to generate class-level weights. This process does not introduce any extra computational cost compared to [23]. We keep the notations of $\boldsymbol{\theta}_\tau$ and $\phi_\tau$ unchanged.

The first stage is embedded in **Step 1**. Explicitly, we obtain the first-stage class-level weights $\boldsymbol{\omega}_i = \omega_i \mathbf{1}$, by cloning the output of the weighting network for $C$ times. Then we leverage the cloned weights $\boldsymbol{\omega}_i$ to manipulate gradients and update $\boldsymbol{\theta}$ with a mini-batch of training instances:

$$\hat{\boldsymbol{\theta}}_\tau(\phi_{\tau-1}) \leftarrow \boldsymbol{\theta}_{\tau-1} - \eta_{\boldsymbol{\theta}} \frac{1}{n} \sum_{i=1}^{n} f_{\boldsymbol{\omega}_i(\phi_{\tau-1})} \left( \nabla_{\boldsymbol{\theta}} l_{train}(x_i, y_i; \boldsymbol{\theta}_{\tau-1}) \right), \tag{9}$$

where $n$ is the mini-batch size, $\eta_{\boldsymbol{\theta}}$ is the learning rate of $\boldsymbol{\theta}$, and $f_{\boldsymbol{\omega}_i(\phi_{\tau-1})}(\cdot)$ is the gradient manipulation operation defined in Eq. (2).

The second stage is embedded in **Step 2** and **Step 3**. Specifically in **Step 2**, GDW optimizes $\phi$ with a mini-batch meta set:

$$\phi_\tau \leftarrow \phi_{\tau-1} - \eta_{\phi} \frac{1}{m} \sum_{i=1}^{m} \nabla_{\phi_{\tau-1}} l_{meta}(x_i^v, y_i^v; \hat{\boldsymbol{\theta}}_\tau(\phi_{\tau-1})), \tag{10}$$

where $m$ is the mini-batch size and $\eta_{\phi}$ is the learning rate of $\phi$. During the back-propagation in updating $\phi_\tau$, GDW generates the second-stage weights using the intermediate gradients $\mathbf{g}_i$ on $\boldsymbol{\omega}_i$.

Note that $\mathbf{g_i} = \frac{1}{m} \sum_{i=1}^m \nabla_{\omega_i} l_{meta}(x_i^v, y_i^v; \hat{\boldsymbol{\theta}}_\tau(\phi_{\tau-1}))$, and we have

$$\boldsymbol{\omega}_i' = \boldsymbol{\omega}_i - \text{clip}(\eta_{\boldsymbol{\omega}} \frac{\mathbf{g_i}}{\|\mathbf{g}\|_1}, -c, c), \tag{11}$$

where $\|\mathbf{g}\|_1$ denotes the $\ell^1$ norm of class-level gradients within a minibatch and $c = 0.2$ represents the clip parameter. Then we impose the zero-mean constraint proposed in Eq. (4) on $\boldsymbol{\omega}_i'$, which is later used in **Step 3** to update $\boldsymbol{\theta}_{\tau-1}$. Note that the two-stage weight generation scheme does not introduce any extra computational cost compared to MWNet because this generation process only utilizes the intermediate gradients during the back-propagation. In **Step 3**, we use $\boldsymbol{\omega}_i'$ to manipulate gradients and update the model parameters $\boldsymbol{\theta}_{\tau-1}$:

$$\boldsymbol{\theta}_\tau \leftarrow \boldsymbol{\theta}_{\tau-1} - \eta_{\boldsymbol{\theta}} \frac{1}{n} \sum_{i=1}^n f_{\boldsymbol{\omega}_i'} \left( \nabla_{\boldsymbol{\theta}} l_{train}(x_i, y_i; \boldsymbol{\theta}_{\tau-1}) \right). \tag{12}$$

The only difference between **Step 1** and **Step 3** is that we use $\boldsymbol{\omega}_i'$ instead of the cloned output of the weighting network $\boldsymbol{\omega}_i$ to optimize $\boldsymbol{\theta}$. Since we only introduce $\phi$ as extra learnable parameters, GDW can scale to large datasets. We summarize GDW in Algorithm 1. Moreover, we visualize the two-stage weight generation process in Figure 2 for better demonstration.

---

**Algorithm 1** Generalized Data Weighting via Class-Level Gradients Manipulation

---

    **Input:** Training set: $D_{train}$, Meta set: $D_{meta}$, batch size $n, m$, # of iterations $T$
    Initial model parameters: $\boldsymbol{\theta}_0$, initial weighting network parameters: $\phi_0$
    **Output:** Trained model: $\boldsymbol{\theta}_T$
1  **for** $\tau \leftarrow 1$ **to** $T$ **do**
2      $\{x_i, y_i\}_{i=1}^n \leftarrow \text{SampleFrom}(D_{train})$
3      $\{x_i^v, y_i^v\}_{i=1}^m \leftarrow \text{SampleFrom}(D_{meta})$
4      Generate $\boldsymbol{\omega}_i$ from $\mathcal{L}_i$ via the weighting network parameterized by $\phi_{\tau-1}$
5      Manipulate gradients by Eq. (2) and update $\hat{\boldsymbol{\theta}}_\tau$ by Eq. (9)
6      Update $\phi_\tau$ by Eq. (10);
7      Update $\boldsymbol{\omega}_i$ to $\boldsymbol{\omega}_i'$ by Eq. (11) and constrain $\boldsymbol{\omega}_i'$ by Eq. (4)
8      Manipulate gradients with $\boldsymbol{\omega}_i'$ by Eq. (2) and update $\boldsymbol{\theta}_\tau$ by Eq. (12)

---

## 4 Experiments

We conduct extensive experiments on classification tasks to examine the performance of GDW. We compare GDW with other methods in the label noise setting and class imbalance setting in Section 4.1 and Section 4.2, respectively. Next, we perform experiments on the real-world dataset Clothing1M [4] in Section 4.3. We conduct further experiments to verify the performance of GDW in the mixed setting, i.e. the coexistence of label noise and class imbalance (see Appendix F for details).

### 4.1 Label Noise Setting

**Setup.** Following [23], we study two settings of label noise: a) Uniform noise: every instance's label uniformly flips to other class labels with probability $p$; b) Flip noise: each class randomly flips to another class with probability $p$. Note that the probability $p$ represents the noise ratio. We randomly select 100 clean images per class from CIFAR10 [47] as the meta set (1000 images in total). Similarly, we select a total of 1000 images from CIFAR100 as its meta set. We use ResNet-32 [48] as the classifier model.

**Comparison methods.** We mainly compare GDW with meta-learning methods: 1) L2RW [22], which assigns weights to instances based on gradient directions; 2) INSW [24], which derives instance weights adaptively from the meta set; 3) MWNet [23]; 4) Soft-label [42], which learns a label smoothing parameter for every instance; 5) Gen-label [40], which generates a meta-soft-label for every instance. We also compare GDW with some traditional methods: 6) BaseModel, which trains ResNet-32 on the noisy training set; 7) Fine-tuning, which uses the meta set to fine-tune the trained model in BaseModel; 8) Co-teaching [20]; 9) GLC [19].

Table 2: Test accuracy on CIFAR10 and CIFAR100 with different uniform noise ratios.

| Dataset | CIFAR10 | | | CIFAR100 | | |
|---|---|---|---|---|---|---|
| | 0% | 40% | 60% | 0% | 40% | 60% |
| BaseModel | $92.73 \pm 0.37$ | $84.38 \pm 0.32$ | $77.92 \pm 0.29$ | $70.42 \pm 0.54$ | $57.28 \pm 0.80$ | $46.86 \pm 1.54$ |
| Fine-tuning | $92.77 \pm 0.37$ | $84.73 \pm 0.47$ | $78.41 \pm 0.31$ | $70.52 \pm 0.57$ | $57.38 \pm 0.87$ | $47.06 \pm 1.47$ |
| Co-teaching | $91.54 \pm 0.39$ | $85.26 \pm 0.56$ | $78.90 \pm 6.64$ | $68.33 \pm 0.13$ | $\underline{59.58 \pm 0.83}$ | $37.74 \pm 2.60$ |
| GLC | $90.85 \pm 0.22$ | $86.12 \pm 0.54$ | $\underline{81.55 \pm 0.60}$ | $65.05 \pm 0.59$ | $56.99 \pm 0.82$ | $41.74 \pm 1.98$ |
| L2RW | $89.70 \pm 0.50$ | $84.66 \pm 1.21$ | $79.98 \pm 1.18$ | $63.40 \pm 1.31$ | $47.06 \pm 4.84$ | $36.02 \pm 2.17$ |
| INSW | $92.70 \pm 0.57$ | $84.88 \pm 0.64$ | $78.77 \pm 0.82$ | $70.52 \pm 0.39$ | $57.11 \pm 0.66$ | $48.00 \pm 1.16$ |
| MWNet | $\mathbf{92.95 \pm 0.33}$ | $86.46 \pm 0.31$ | $81.14 \pm 0.94$ | $\mathbf{70.64 \pm 0.31}$ | $58.37 \pm 0.33$ | $\underline{50.21 \pm 2.98}$ |
| Soft-label | $92.63 \pm 0.27$ | $\underline{86.52 \pm 0.10}$ | $80.94 \pm 0.25$ | $70.50 \pm 0.44$ | $57.48 \pm 0.43$ | $48.18 \pm 0.89$ |
| Gen-label | $92.56 \pm 0.56$ | $84.68 \pm 0.57$ | $78.32 \pm 0.94$ | $70.46 \pm 0.37$ | $57.86 \pm 0.50$ | $48.08 \pm 0.98$ |
| **GDW** | $\underline{92.94 \pm 0.15}$ | $\mathbf{88.14 \pm 0.35}$ | $\mathbf{84.11 \pm 0.21}$ | $\underline{70.65 \pm 0.52}$ | $\mathbf{59.82 \pm 1.62}$ | $\mathbf{53.33 \pm 3.70}$ |

Table 3: Test accuracy on CIFAR10 and CIFAR100 with different flip noise ratios.

| Dataset | CIFAR10 | | | CIFAR100 | | |
|---|---|---|---|---|---|---|
| | 0% | 20% | 40% | 0% | 20% | 40% |
| BaseModel | $92.73 \pm 0.37$ | $90.14 \pm 0.35$ | $81.20 \pm 0.93$ | $70.42 \pm 0.54$ | $64.96 \pm 0.16$ | $49.83 \pm 0.82$ |
| Fine-tuning | $92.77 \pm 0.37$ | $90.15 \pm 0.36$ | $81.53 \pm 0.96$ | $70.52 \pm 0.57$ | $65.02 \pm 0.22$ | $50.23 \pm 0.71$ |
| Co-teaching | $91.54 \pm 0.39$ | $89.27 \pm 0.24$ | $69.77 \pm 3.97$ | $68.33 \pm 0.13$ | $62.96 \pm 0.73$ | $42.54 \pm 1.68$ |
| GLC | $90.85 \pm 0.22$ | $\underline{90.22 \pm 0.13}$ | $\mathbf{89.74 \pm 0.19}$ | $65.05 \pm 0.59$ | $64.11 \pm 0.40$ | $\mathbf{63.11 \pm 0.93}$ |
| L2RW | $89.70 \pm 0.50$ | $88.21 \pm 0.49$ | $82.90 \pm 1.27$ | $63.40 \pm 1.31$ | $55.27 \pm 2.27$ | $45.41 \pm 2.53$ |
| INSW | $92.70 \pm 0.57$ | $89.90 \pm 0.45$ | $80.09 \pm 2.00$ | $70.52 \pm 0.39$ | $\underline{65.32 \pm 0.27}$ | $50.13 \pm 0.39$ |
| MWNet | $\mathbf{92.95 \pm 0.33}$ | $89.93 \pm 0.17$ | $85.55 \pm 0.82$ | $\mathbf{70.64 \pm 0.31}$ | $64.72 \pm 0.68$ | $50.62 \pm 0.46$ |
| Soft-label | $92.63 \pm 0.27$ | $90.17 \pm 0.47$ | $85.52 \pm 0.78$ | $70.50 \pm 0.44$ | $65.20 \pm 0.45$ | $50.97 \pm 0.41$ |
| Gen-label | $92.56 \pm 0.56$ | $90.18 \pm 0.13$ | $80.93 \pm 1.29$ | $70.46 \pm 0.37$ | $64.94 \pm 0.53$ | $49.93 \pm 0.55$ |
| **GDW** | $\underline{92.94 \pm 0.15}$ | $\mathbf{91.05 \pm 0.26}$ | $\underline{87.70 \pm 0.37}$ | $\underline{70.65 \pm 0.52}$ | $\mathbf{65.41 \pm 0.75}$ | $\underline{52.44 \pm 0.79}$ |

**Training.** Most of our training settings follow [23] and we use the cosine learning rate decay schedule [49] for a total of 80 epochs for all methods. See Appendix C for details.

**Analysis.** For all experiments, we report the mean and standard deviation over 5 runs in Table 2 and Table 3, where the best results are in **bold** and the second-best results are marked by underlines. First, we can observe that GDW outperforms nearly all the competing methods in all noise settings except for the 40% flip noise setting. Under this setting, GLC estimates the label corruption matrix well and thus performs the best, whereas the flip noise assumption scarcely holds in real-world scenarios. Note that GLC also performs much better than MWNet under the 40% flip noise setting as reported in [23]. Besides, under all noise settings, GDW has a consistent performance gain compared with MWNet, which aligns with our motivation in Figure 1. Furthermore, as the ratio increases from 40% to 60% in the uniform noise setting, the gap between GDW and MWNet increases from 1.68% to 2.97% in CIFAR10 and 1.45% to 3.12% in CIFAR100. Even under 60% uniform noise, GDW still has low test errors in both datasets and achieves more than 3% gain in CIFAR10 and 6% gain in CIFAR100 compared with the second-best method. Last but not least, GDW outperforms Soft-label and Gen-label in all settings. One possible reason is that manipulating gradient flows is a more direct way to capture class-level information than learning labels.

In Figure 3, we show the distribution of class-level target weight ($\omega_t$) on clean and noisy instances in one epoch under the CIFAR10 40% uniform noise setting. We observe that $\omega_t$ of most clean instances are larger than that of most noisy instances, which indicates that $\omega_t$ can distinguish between clean instances and noisy instances. This is consistent with Eq. (3) that $\omega_t$ serves as the instance weight.

To better understand the changing trend of non-target class-level weights, we visualize the ratio of increased weights in one epoch in Figure 5 under the CIFAR10 40% uniform noise setting. Specifically, there are three categories: **n**on-**t**arget **w**eights on **c**lean instances ($\omega_{nt}^c$), **t**rue **t**arget **w**eights on **n**oisy instances ($\omega_{tt}^n$) and **n**on-**t**arget (excluding true targets) **w**eights on **n**oisy instances ($\omega_{nt}^n$). Formally, "target weight" means the class-level weight on the label position. "true-target weight" means the class-level weight on the true label position, which are only applicable for noisy instances. "non-target weight" means the class-level weight except the label position and the true label position. For example, as shown in Figure 1 where a cat is mislabeled as "dog", the corresponding meanings of the notations are as follows: 1) $\omega_t^n$ means $\omega_{dog}$ ("dog" is the target); 2) $\omega_{tt}^n$ means $\omega_{cat}$ ("cat" is the ture target); 3) $\omega_{nt}^n$ means $\omega_{bird}$ ("bird" is one of the non-targets). For a correctly labeled

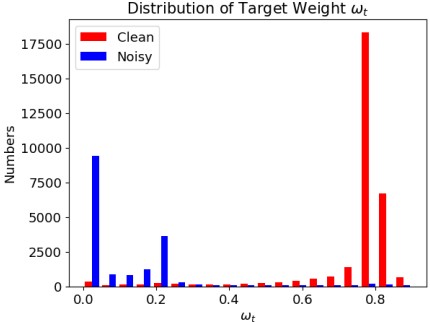

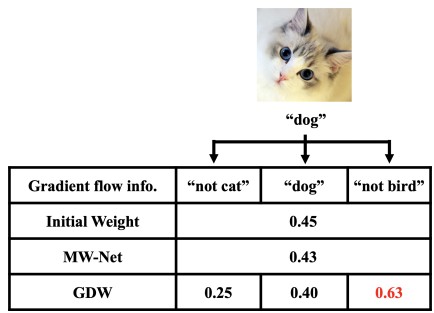

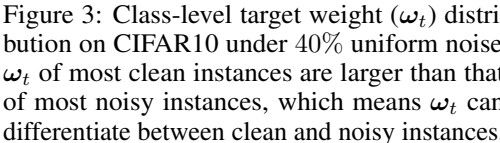

Figure 3: Class-level target weight ($\boldsymbol{\omega}_t$) distribution on CIFAR10 under $40\%$ uniform noise. $\boldsymbol{\omega}_t$ of most clean instances are larger than that of most noisy instances, which means $\boldsymbol{\omega}_t$ can differentiate between clean and noisy instances.

Figure 4: The change of class-level weights in an iteration for a noisy instance (cat mislabeled as "dog"). MWNet downweights all gradient flows. In contrast, GDW upweights the "not bird" gradient flow for better information use.

cat, the corresponding meanings are: 1) $\boldsymbol{\omega}_t^c = \boldsymbol{\omega}_{tt}^c$ means $\boldsymbol{\omega}_{cat}$ ("cat" is both the target and the ture target); 2) $\boldsymbol{\omega}_{nt}^c$ means $\boldsymbol{\omega}_{dog}$ and $\boldsymbol{\omega}_{bird}$ ("dog" and "bird" are both non-targets).

Note that in Figure 1, $\boldsymbol{\omega}_{tt}^n$ represents the importance of the "not cat" gradient flow and $\boldsymbol{\omega}_{nt}^n$ represents the importance of the "not bird" gradient flow. If the cat image in Figure 1 is correctly labeled as "cat", then the two non-target weights $\boldsymbol{\omega}_{nt}^c$ are used to represent the importance of the "not dog" and the "not bird" gradient flows, respectively. In one epoch, we calculate **the ratios of** the number of increased $\boldsymbol{\omega}_{nt}^c$, $\boldsymbol{\omega}_{tt}^n$ and $\boldsymbol{\omega}_{nt}^n$ **to** the number of all corresponding weights. $\boldsymbol{\omega}_{nt}^c$ and $\boldsymbol{\omega}_{nt}^n$ are expected to increase since their gradient flows contain valuable information, whereas $\boldsymbol{\omega}_{tt}^n$ is expected to decrease because the "not cat" gradient flow contains harmful information. Figure 5 aligns perfectly with our expectation. Note that the lines of $\boldsymbol{\omega}_{nt}^c$ and $\boldsymbol{\omega}_{nt}^n$ nearly coincide with each other and fluctuate around $65\%$. This means non-target weights on clean instances and noisy instances share the same changing pattern, i.e., around $65\%$ of $\boldsymbol{\omega}_{nt}^c$ and $\boldsymbol{\omega}_{nt}^n$ increase. Besides, less than $20\%$ of $\boldsymbol{\omega}_{tt}^n$ increase and thus more than $80\%$ decrease, which means the gradient flows of $\boldsymbol{\omega}_{tt}^n$ contain much harmful information.

In Figure 4, we show the change of class-level weights in an iteration for a noisy instance, i.e., a cat image mislabeled as "dog". The gradient flows of "not cat" and "dog" contain harmful information and thus are downweighted by GDW. In addition, GDW upweights the valuable "not bird" gradient flow from 0.45 to 0.63. By contrast, unable to capture class-level information, MWNet downweights all gradient flows from 0.45 to 0.43, which leads to information loss on the "not bird" gradient flow.

**Training without the zero-mean constraint.** We have also tried training without the zero-mean constraint in Section 3.3 and got poor results (see Appendix B.2 for details). Denote the **t**rue **t**arget as $tt$ and one of the **n**on-**t**arget labels as $nt$ ($nt \neq tt$). Note that the gradient can be unrolled as (see Appendix B.2 for details):

$$f_{\boldsymbol{\omega}}\left(\nabla_{\boldsymbol{\theta}}\mathcal{L}\right) = \boldsymbol{\omega}_t \sum_j \left(\boldsymbol{p}_j' - \mathbf{y}_j\right)\frac{\partial \boldsymbol{l}_j}{\partial \boldsymbol{\theta}} + \left(\sum_k \boldsymbol{\omega}_k \boldsymbol{p}_k - \boldsymbol{\omega}_t\right)\sum_j \boldsymbol{p}_j' \frac{\partial \boldsymbol{l}_j}{\partial \boldsymbol{\theta}}. \tag{13}$$

If $\sum_k \boldsymbol{\omega}_k \boldsymbol{p}_k - \boldsymbol{\omega}_t$ is positive and the learning rate is small enough, $\left(\sum_k \boldsymbol{\omega}_k \boldsymbol{p}_k - \boldsymbol{\omega}_t\right)\boldsymbol{p}_{tt}'\frac{\partial \boldsymbol{l}_{tt}}{\partial \boldsymbol{\theta}}$ contributes to the decrease of the true target logit $\boldsymbol{l}_{tt}$ after a gradient descent step. If negative, $\left(\sum_k \boldsymbol{\omega}_k \boldsymbol{p}_k - \boldsymbol{\omega}_t\right)\boldsymbol{p}_{nt}'\frac{\partial \boldsymbol{l}_{nt}}{\partial \boldsymbol{\theta}}$ contributes to the increase of the non-target logit $\boldsymbol{l}_{nt}$. Therefore, without the zero-mean constraint, the second term in Eq. (13) may hurt the performance of the model regardless of the sign of $\sum_k \boldsymbol{\omega}_k \boldsymbol{p}_k - \boldsymbol{\omega}_t$. Similarly, training without the constraint results in poor performance in other settings. Hence we omit those results in the following subsections.

## 4.2 Class Imbalance Setting

**Setup and comparison methods.** The imbalance factor $\mu \in (0, 1)$ of a dataset is defined as the number of instances in the smallest class divided by that of the largest [23]. Long-Tailed CIFAR [47] are created by reducing the number of training instances per class according to an exponential function $n = n_i \mu^{i/(C-1)}$, where $i$ is the class index (0-indexed) and $n_i$ is the original number of

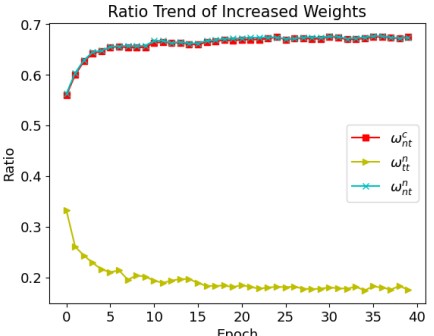
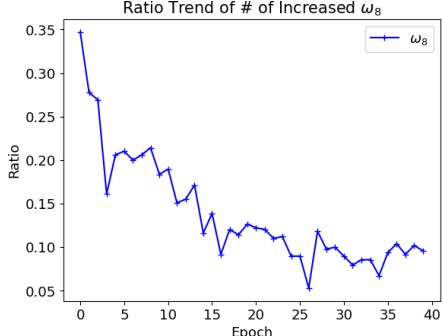

Figure 5: Ratio trend of the number of increased $\omega_{nt}^c$, $\omega_{tt}^n$, and $\omega_{nt}^n$ under the CIFAR10 40% uniform noise setting. Around 65% of $\omega_{nt}^c$ and $\omega_{nt}^n$ increase since they contain useful information. Besides, less than 20% of $\omega_{tt}^n$ increase and thus more than 80% of $\omega_{tt}^n$ decrease since they contain harmful information.

Figure 6: Ratio trend of the number of increased $\omega_8$ on $C9$ instances under the Long-Tailed CIFAR10 $\mu = 0.1$ setting. Less than 10% of $\omega_8$ increase and thus more than 90% decrease. A small $\omega_8$ strikes a balance between two kinds of information: "$C8$" and "not $C8$", which better handles class imbalance.

Table 4: Test accuracy on the long-tailed CIFAR10 and CIFAR100 with different imbalance ratios.

| Dataset | CIFAR10 | | | CIFAR100 | | |
|---|---|---|---|---|---|---|
| | $\mu = 1$ | $\mu = 0.1$ | $\mu = 0.01$ | $\mu = 1$ | $\mu = 0.1$ | $\mu = 0.01$ |
| BaseModel | $92.73 \pm 0.37$ | $85.93 \pm 0.57$ | $69.77 \pm 1.13$ | $70.42 \pm 0.54$ | $56.25 \pm 0.49$ | $37.79 \pm 0.82$ |
| Fine-tuning | $92.77 \pm 0.37$ | $82.60 \pm 0.49$ | $59.76 \pm 1.00$ | $70.52 \pm 0.57$ | $55.95 \pm 0.50$ | $37.10 \pm 0.87$ |
| Focal | $91.68 \pm 0.49$ | $84.57 \pm 0.83$ | $65.78 \pm 4.02$ | $68.48 \pm 0.38$ | $55.02 \pm 0.51$ | $37.43 \pm 1.00$ |
| Balanced | $92.80 \pm 0.47$ | $86.05 \pm 0.46$ | $63.63 \pm 3.60$ | $70.56 \pm 0.56$ | $55.02 \pm 0.80$ | $27.60 \pm 1.39$ |
| L2RW | $89.70 \pm 0.50$ | $79.11 \pm 3.40$ | $51.15 \pm 7.13$ | $63.40 \pm 1.31$ | $46.28 \pm 4.51$ | $25.86 \pm 5.78$ |
| INSW | $92.70 \pm 0.57$ | $\underline{86.31 \pm 0.28}$ | $\underline{70.27 \pm 0.24}$ | $70.52 \pm 0.39$ | $55.94 \pm 0.51$ | $37.67 \pm 0.59$ |
| MWNet | $\mathbf{92.95 \pm 0.33}$ | $86.17 \pm 0.75$ | $62.70 \pm 1.76$ | $\underline{\mathbf{70.64 \pm 0.31}}$ | $\underline{56.49 \pm 1.52}$ | $\underline{37.83 \pm 0.86}$ |
| **GDW** | $\underline{\mathbf{92.94 \pm 0.15}}$ | $\mathbf{86.77 \pm 0.55}$ | $\mathbf{71.31 \pm 1.03}$ | $\mathbf{70.65 \pm 0.52}$ | $\mathbf{56.78 \pm 0.52}$ | $\mathbf{37.94 \pm 1.58}$ |

training instances. Comparison methods include: 1) L2RW [22]; 2) INSW [24]; 3) MWNet [23]; 4) BaseModel; 5) Fine-tuning; 6) Balanced [13]; 7) Focal [21].

**Analysis.** As shown in Table 4, GDW performs best in nearly all settings and exceeds MWNet by 8.6% when the imbalance ratio $\mu$ is 0.01 in CIFAR10. Besides, INSW achieves competitive performance at the cost of introducing a huge amount of learnable parameters (equal to the training dataset size $N$). Furthermore, we find that BaseModel achieves competitive performance, but fine-tuning on the meta set hurts the model's performance. We have tried different learning rates from $10^{-7}$ to $10^{-1}$ for fine-tuning, but the results are similar. One explanation is that the balanced meta set worsens the model learned from the imbalanced training set. These results align with the experimental results in [24] which also deals with class imbalance.

Denote the smallest class as $C9$ and the second smallest class as $C8$ in Long-Tailed CIFAR10 with $\mu = 0.1$. Recall that $\omega_j$ denotes the $j^{th}$ class-level weight. For all $C9$ instances in an epoch, we calculate **the ratio of** the number of increased $\omega_8$ **to** the number of all $\omega_8$, and then visualize the ratio trend in Figure 6. Since $C9$ is the smallest class, instance weighting methods upweight both $\omega_8$ and $\omega_9$ on a $C9$ instance. Yet in Figure 6, less than 10% of $\omega_8$ increase and thus more than 90% decrease. This can be explained as follows. There are two kinds of information in the long-tailed dataset regarded to $C8$: "$C8$" and "not $C8$". Since $C8$ belongs to the minority class, the dataset is biased towards the "not $C8$" information. Because $\omega_8$ represents the importance of "not $C8$", a smaller $\omega_8$ weakens the "not $C8$" information. As a result, decreased $\omega_8$ achieves a balance between two kinds of information: "$C8$" and "not $C8$", thus better handling class imbalance at the class level. We have conducted further experiments on imbalanced settings to verify the effectiveness of GDW and see Appendix D for details.

Table 5: Test accuracy on Clothing1M.

| Method | BaseModel | Fine-tuning | Co-teaching | GLC | L2RW | INSW | MWNet | Soft-label | Gen-label | **GDW** |
|---|---|---|---|---|---|---|---|---|---|---|
| Accuracy(%) | 65.02 | 67.68 | 68.13 | 68.60 | 68.80 | 68.25 | 68.46 | 68.69 | 67.64 | **69.39** |

## 4.3 Real-world Setting

**Setup and training.** The Clothing1M dataset contains one million images from fourteen classes collected from the web [4]. Labels are constructed from surrounding texts of images and thus contain some errors. We use the ResNet-18 model pre-trained on ImageNet [50] as the classifier. The comparison methods are the same as those in the label noise setting since the main issue of Clothing1M is label noise [4]. All methods are trained for 5 epochs via SGD with a $0.9$ momentum, a $10^{-3}$ initial learning rate, a $10^{-3}$ weight decay, and a $128$ batchsize. See Appendix E for details.

**Analysis.** As shown in Table 5, GDW achieves the best performance among all the comparison methods and outperforms MWNet by $0.93\%$. In contrast to unsatisfying results in previous settings, L2RW performs quite well in this setting. One possible explanation is that, compared with INSW and MWNet which update weights iteratively, L2RW obtains instance weights only based on current gradients. As a result, L2RW can more quickly adapt to the model's state, but meanwhile suffers from unstable weights [23]. In previous settings, we train models from scratch, which need stable weights to stabilize training. Therefore, INSW and MWNet generally achieve better performance than L2RW. Whereas in this setting, we use the pre-trained ResNet-18 model which is already stable enough. Thus L2RW performs better than INSW and MWNet.

## 5 Conclusion

Many instance weighting methods have recently been proposed to tackle label noise and class imbalance, but they cannot capture class-level information. For better information use when handling the two issues, we propose GDW to generalize data weighting from instance level to class level by reweighting gradient flows. Besides, to efficiently obtain class-level weights, we design a two-stage weight generation scheme which is embedded in a three-step bi-level optimization framework and leverages intermediate gradients to update class-level weights via a gradient descent step. In this way, GDW achieves remarkable performance improvement in various settings.

The limitations of GDW are two-fold. Firstly, the gradient manipulation is only applicable to single-label classification tasks. When applied to multi-label tasks, the formulation of gradient manipulation need some modifications. Secondly, GDW does not outperform comparison methods by a large margin in class imbalance settings despite the potential effectiveness analyzed in Section 4.2. One possible explanation is that better information utilization may not result in performance gain which also depends on various other factors.

## 6 Acknowledgement

We thank Prof. Hao Zhang from Tsinghua University for helpful suggestions. This research was supported in part by the MSR-Mila collaboration funding. Besides, this research was empowered in part by the computational support provided by Compute Canada (www.computecanada.ca).

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
