# A Derivation of D1

Denote the logit vector as $\boldsymbol{x}$, we have

$$\boldsymbol{p}_j = \frac{e^{\boldsymbol{x}_j}}{\sum_k e^{\boldsymbol{x}_k}} \tag{14}$$

$$\mathcal{L} = -\sum_j \boldsymbol{y}_j \log \boldsymbol{p}_j. \tag{15}$$

For the target (label) position $t$ we have $\boldsymbol{y}_t = 1$ and

$$\frac{\partial \mathcal{L}}{\partial \boldsymbol{x}_t} = \frac{\partial \mathcal{L}}{\partial \boldsymbol{p}_t} \frac{\partial \boldsymbol{p}_t}{\partial \boldsymbol{x}_t} = -\frac{\sum_k e^{\boldsymbol{x}_k}}{e^{\boldsymbol{x}_t}} \frac{e^{\boldsymbol{x}_t} \sum_k e^{\boldsymbol{x}_k} - e^{2\boldsymbol{x}_t}}{\left(\sum_k e^{\boldsymbol{x}_k}\right)^2} = \frac{e^{\boldsymbol{x}_t}}{\sum_k e^{\boldsymbol{x}_k}} - 1 = \boldsymbol{p}_t - \boldsymbol{y}_t. \tag{16}$$

For any other position $j$ ($j \neq t$), we have $\boldsymbol{y}_j = 0$ and

$$\frac{\partial \mathcal{L}}{\partial \boldsymbol{x}_j} = \frac{\partial \mathcal{L}}{\partial \boldsymbol{p}_t} \frac{\partial \boldsymbol{p}_t}{\partial \boldsymbol{x}_j} = -\frac{\sum_k e^{\boldsymbol{x}_k}}{e^{\boldsymbol{x}_t}} \frac{-e^{\boldsymbol{x}_j + \boldsymbol{x}_t}}{\left(\sum_k e^{\boldsymbol{x}_k}\right)^2} = \frac{e^{\boldsymbol{x}_j}}{\sum_k e^{\boldsymbol{x}_k}} = \boldsymbol{p}_j - \boldsymbol{y}_j. \tag{17}$$

Therefore, we can conclude that $\boldsymbol{D}_1 = \boldsymbol{p} - \boldsymbol{y}$.

# B Zero-mean Constraint on Class-level Weights

## B.1 Derivation

$$\boldsymbol{\omega}_j(\boldsymbol{p}_j - \mathbf{y}_j) = \boldsymbol{\omega}_j \boldsymbol{p}_j - \boldsymbol{\omega}_j \mathbf{y}_j \tag{18}$$

$$= \left(\sum_k \boldsymbol{\omega}_k \boldsymbol{p}_k\right) \frac{\boldsymbol{\omega}_j \boldsymbol{p}_j}{\sum_k \boldsymbol{\omega}_k \boldsymbol{p}_k} - \left(\sum_k \boldsymbol{\omega}_k \mathbf{y}_k\right) \frac{\boldsymbol{\omega}_j \mathbf{y}_j}{\sum_k \boldsymbol{\omega}_k \mathbf{y}_k} \tag{19}$$

If $\mathbf{y}_j = 0$, the second term of (19) becomes 0, therefore can be rewritten as $\boldsymbol{\omega}_t \mathbf{y}_j$

$$= \left(\sum_k \boldsymbol{\omega}_k \boldsymbol{p}_k\right) \boldsymbol{p}'_j - \boldsymbol{\omega}_t \mathbf{y}_j \tag{20}$$

$$= \boldsymbol{\omega}_t \left(\boldsymbol{p}'_j - \mathbf{y}_j\right) + \left(\sum_k \boldsymbol{\omega}_k \boldsymbol{p}_k - \boldsymbol{\omega}_t\right) \boldsymbol{p}'_j. \tag{21}$$

## B.2 Training without Zero-mean Constraint

$$f_{\boldsymbol{\omega}}(\nabla_{\boldsymbol{\theta}} \mathcal{L}) = f_{\boldsymbol{\omega}}(\sum_j \frac{\partial \mathcal{L}_i}{\partial \boldsymbol{l}_j} \frac{\partial \boldsymbol{l}_j}{\partial \boldsymbol{\theta}})$$

$$= \sum_j \boldsymbol{\omega}_j (\boldsymbol{p}_j - \mathbf{y}_j) \frac{\partial \boldsymbol{l}_j}{\partial \boldsymbol{\theta}} \tag{22}$$

$$= \boldsymbol{\omega}_t \sum_j (\boldsymbol{p}'_j - \mathbf{y}_j) \frac{\partial \boldsymbol{l}_j}{\partial \boldsymbol{\theta}} + (\sum_k \boldsymbol{\omega}_k \boldsymbol{p}_k - \boldsymbol{\omega}_t) \sum_j \boldsymbol{p}'_j \frac{\partial \boldsymbol{l}_j}{\partial \boldsymbol{\theta}}.$$

Without zero-mean constraint, the training becomes unstable. We plot the training curve of the CIFAR10 40% uniform noise setting in Figure 7.

# C Label Noise Training Setting

Following the training setting of [23], the classifier network is trained with SGD with a weight decay 5e-4, an initial learning rate of 1e-1 and a mini-batch size of 100 for all methods. We use the cosine learning rate decay schedule [49] for a total of 80 epochs. We set the outer level learning $\eta_{\boldsymbol{\omega}}$ as

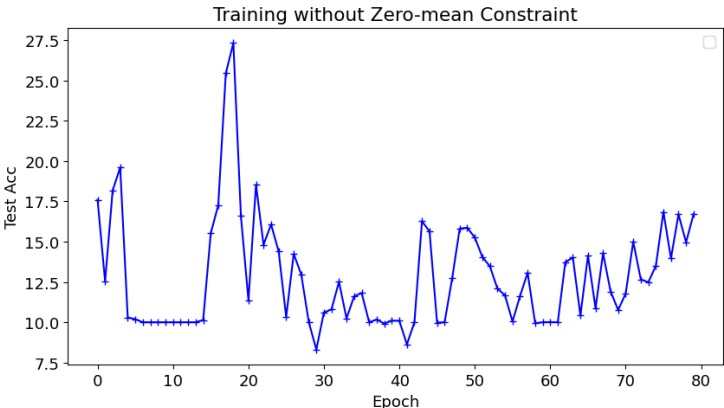

Figure 7: Training curve without zero-mean constraint on CIFAR10 under 40% uniform noise.

100 on CIFAR10 and as 1000 on CIFAR100. The MLP weighting network is trained with Adam [51] with a fixed learning rate 1e-3 and a weight decay 1e-4. For GLC, we first train 40 epochs to estimate the label corruption matrix and then train another 40 epochs to evaluate its performance. Since Co-teach uses two models, each model is trained for 40 epochs for a fair comparison. We repeat every experiment 5 times with different random seeds (seed=1, 10, 100, 1000, 10000, respectively) for network initialization and label noise generation. We report the average test accuracy over the last 5 epochs as the model's final performance. We use one V100 GPU for all the experiments.

## D  Further Experiments on Imbalanced Setting

### D.1  Class-level Weight Analysis

We conduct one more experiment under the imbalance setting to better verify the interpretability of GDW. As shown in Table 6, we report the ratio of the number of increased $\omega_i$ after gradient update on $C_j$ instances in one epoch, where $C_0$ is the largest class and $C_9$ is the smallest class.

Note that $\omega_i$ on $C_i$ contains the "is $C_i$" information in the dataset. As a result, $\omega_i$ on $C_i$ should be large for small classes and small for large classes. As shown above, the ratio of increased $\omega_i$ on $C_i$ (the diagonal elements) increases from $0.036$ to $0.935$ as $i$ increases from 0 to 9.

On the other hand, $\omega_i$ on $C_j$ ($j \neq i$) contains the "not $C_i$" information in the dataset. If $i$ is a large class, $\omega_i$ on $C_j$ ($j \neq i$) should be large and vice versa. For $\omega_i$ ($i = 0, 1, 2, 3, 4$), the ratio of increased $\omega_i$ on $C_j$ ($j \neq i$) are generally larger than $0.5$, and for $\omega_i$ ($i = 5, 6, 7, 8, 9$), the ratio of increased $\omega_i$ on $C_j$ ($j \neq i$) are generally less than $0.5$. These results align with our analysis on the interpretable information of gradient flows.

Table 6: Ratio of increased class-level weights under the imbalance setting.

| weight/class | $C_0$ | $C_1$ | $C_2$ | $C_3$ | $C_4$ | $C_5$ | $C_6$ | $C_7$ | $C_8$ | $C_9$ |
|---|---|---|---|---|---|---|---|---|---|---|
| $\omega_0$ | 0.036 | 0.968 | 0.973 | 0.972 | 0.965 | 0.974 | 0.972 | 0.976 | 0.956 | 0.973 |
| $\omega_1$ | 0.887 | 0.095 | 0.912 | 0.929 | 0.907 | 0.927 | 0.911 | 0.922 | 0.910 | 0.920 |
| $\omega_2$ | 0.848 | 0.844 | 0.141 | 0.839 | 0.822 | 0.845 | 0.818 | 0.847 | 0.829 | 0.802 |
| $\omega_3$ | 0.585 | 0.608 | 0.552 | 0.405 | 0.569 | 0.541 | 0.561 | 0.559 | 0.617 | 0.596 |
| $\omega_4$ | 0.474 | 0.521 | 0.420 | 0.460 | 0.509 | 0.455 | 0.456 | 0.482 | 0.467 | 0.512 |
| $\omega_5$ | 0.291 | 0.261 | 0.288 | 0.252 | 0.309 | 0.701 | 0.303 | 0.267 | 0.297 | 0.257 |
| $\omega_6$ | 0.199 | 0.189 | 0.169 | 0.198 | 0.196 | 0.222 | 0.778 | 0.195 | 0.207 | 0.182 |
| $\omega_7$ | 0.117 | 0.117 | 0.105 | 0.084 | 0.115 | 0.079 | 0.126 | 0.920 | 0.133 | 0.090 |
| $\omega_8$ | 0.115 | 0.124 | 0.178 | 0.185 | 0.184 | 0.174 | 0.191 | 0.181 | 0.862 | 0.137 |
| $\omega_9$ | 0.043 | 0.050 | 0.064 | 0.061 | 0.074 | 0.062 | 0.097 | 0.069 | 0.040 | 0.935 |

### D.2 Experiments on Places-LT

Table 7: Test accuracy on Places-LT.

| Method | L2RW | INSW | MW-Net | **GDW** |
|---|---|---|---|---|
| Accuracy (%) | 15.08 | 17.80 | 18.08 | **19.17** |

We have conducted experiments on the Places-LT dataset [32] and compared GDW with other meta-learning-based methods. For all methods, the weight decay is set to $0.001$ and the batchsize is set to $64$. We adopt a $0.01$ initial learning rate and a cosine learning rate decay policy for 10 epochs. The weight decay is set to $0.001$. The backbone network is ResNet18 and we use the ImageNet pre-trained model for initialization.

As shown in Table 7, GDW achieves the best performance among all the comparison methods and outperforms MWNet by $1.09\%$. This improvement is larger than that of CIFAR100. The reason is that GDW can manipulate class-level information and thus performs better on the dataset with a larger number of classes (365 in Places-LT and 100 in CIFAR100). Besides, we can observe that L2RW performs the worst and the reason may be that L2RW suffers from unstable weights [23].

## E  Real-world Training Setting

Similar to [23], we use the 7k validation set as the meta set and the origin test set to evaluate the classifier's final performance. For GLC, we first train 2 epochs to estimate the label corruption matrix and then train another 3 epochs to evaluate its performance. Since Co-teach uses two models, each model is trained for 3 epochs for a fair comparison.

## F  Experiments on Mixed Setting

We conduct further experiments to verify the performance of GDW in the mixed setting, i.e. the coexistence of label noise and class imbalance. Specifically, we compare GDW with the mostly second-best method MW-Net [23] under the mixed setting of uniform noise and class imbalance on CIFAR10 and CIFAR100. As shown in Table 8, GDW demonstrates great performance gain over MW-Net, which means GDW can simultaneously better tackle both problems.

Table 8: Test accuracy under mixed settings.

| Dataset | Noise Ratio | Imb Factor | MW-Net | GDW |
|---|---|---|---|---|
| CIFAR10 | 0.40 | 0.10 | 71.54 | 76.30 |
| CIFAR10 | 0.60 | 0.10 | 61.62 | 70.24 |
| CIFAR10 | 0.40 | 0.01 | 48.04 | 48.53 |
| CIFAR10 | 0.60 | 0.01 | 39.51 | 40.07 |
| CIFAR100 | 0.40 | 0.10 | 36.10 | 38.20 |
| CIFAR100 | 0.60 | 0.10 | 24.80 | 25.40 |
| CIFAR100 | 0.40 | 0.01 | 21.26 | 22.07 |
| CIFAR100 | 0.60 | 0.01 | 12.75 | 14.15 |