# OpenReview forum: "Generalized DataWeighting via Class-Level Gradient Manipulation"
_NeurIPS.cc/2021/Conference — NeurIPS 2021 Poster_

### Official Review · Reviewer_Khcd · 2021-07-06

**Rating:** 5
**Confidence:** 4

**Summary:**

In the paper, the authors propose a novel generalized data weighting method to improve the performance of existing label noise and class imbalance methods by manipulating the class-level gradients. Existing instance weighting methods are special cases of the proposed method when the class-level weights are the same. The proposed method is quite simple, easy to be deployed to existing methods, and does not introduce additional computational costs.

**Limitations And Societal Impact:**

No, the authors forgot to address the limitations and potential negative societal impact of their work. I suggest adding some discussions on this point.

**Main Review:**

Comments:
1. In equation (8), performing empirical risk minimization on a small meta dataset may be very likely to overfitting the meta dataset.
2. For the experiments, how to ensure a fair comparison with traditional methods such as co-teaching [20] where meta dataset is not needed.
3. In Figure 3, some class-level target weights for noisy data are still very large, even larger than that for the clean data. Giving the mislabeled data large weights could greatly harm the performance. I'm not sure whether the Noisy (the blue in legend) in Figure 3 means a group of intact and mislabeled data, or only mislabeled data.
4. Experiments of the paper include label noise (uniform + flip noise + real-world label noise) and class imbalance experiments. It seems that the proposed method largely outperforms other baselines only for the cases of uniform label noise with large noise ratios. For example, for the flip noise setting in label noise, it is hard to conclude the proposed method is better than others.

Minor comments:
1. The method in [20] is named co-teaching, not co-teach.
2. Class imbalance mainly refers to the setting that both training and test sets are class imbalanced. If the training set is class imbalanced and the test set is class balanced (the setting in the paper I think), better to call it class-prior shift or label shift.

----------------------------POST REBUTTAL-----------------------------

Thank the authors very much for the rebuttal! After reading the rebuttal and all the reviews, the authors have addressed my concerns properly. Unfortunately, I still tend to reject this paper:

1. The paper mentioned in the abstract about a novel problem setting (simultaneously handle label noise and imbalance) but they didn't present any experiment results under the claimed setting in the main paper. The authors should either add more experiments under the claimed setting (more than the experiments they presented in the rebuttal) or change the position of the paper writing.
2. I still have a concern about the effectiveness of the proposed method in experiments. For example, it seems the proposed method works not that well in flip label noise cases. Also, Figure 3 shows the proposed method cannot well differentiate correct and mislabeled data under the uniform noise. I guess the same figure for the flip label noise could be worse.


**Time Spent Reviewing:**

3

---

> ### Author Response · Authors · 2021-08-09
> **Response to review question**
>
> ### 0. General Reply
>
> Many thanks for your valuable and constructive comments on clarifying, correcting, and improving the materials in this paper! We will carefully revise the paper according to your comments as explained below.
>
> ### 1. About Equation (8)
>
> Thanks for your comments. We are not performing ERM directly on the meta-set. The meta-set is only used to optimize the weighting network, rather than the classification model. The purpose of this optimization is to find the optimal class-level weights (or weighting network parameters) at the current classification model parameter $\boldsymbol \theta$. Since we do not use the meta-set to optimize the classification model, no overfitting problem will occur.
>
> ### 2. Fair Comparison
>
> Thanks for the good question. For a fair comparison, we merge the training set and the meta-set for methods not requiring a meta-set. Additionally, we keep the same training setting and hyperparameters.
>
> ### 3. Figure 3
>
> Thanks for the good question. The blue legend Noisy stands for all the mislabeled data.
>
> As shown in **Figure 3**, most noisy data are given small weights while only a very small portion of them are given large weights. This is most probably because of the imperfectness of our algorithm, in that GDW cannot completely differentiate clean and noisy data. However, the impact of the very small portion of noisy data will be cancelled by most correctly weighted data instances. This is validated by the good performance of GDW as shown in **Table 2** and **Table 3**.
>
> ### 4. Effectiveness of GDW
>
> Thanks for the comments.
>
> In the flip noise setting, except that GLC [19] performs better in the 40% setting, GDW always gets the best performance compared with other methods and gets large performance gain in most settings (e.g. 20%, 40% on CIFAR10, and 40% on CIFAR100). In the 20% setting on CIFAR100, GDW gets comparable results compared to the second-best. GLC [19] estimates a label corruption matrix to correct the prediction. This estimation is quite easy under the 40% flip noise (flip one class to another class w.p. 40%). This is why GLC gets better results. However, under uniform noise scenarios, as shown in **Table 2**, GLC yields bad performance compared to GDW. This is because uniform noise is harder to estimate. Moreover, we explained in Line 211-216 that 40% flip noise setting is rare in real-world datasets.
>
> Furthermore, we did comparison experiments under the co-existing setting after receiving this comment. Specifically, we compared with the mostly second-best method Meta-Weight-Net [23] under the mixed setting of uniform noise and class imbalance on CIFAR10 and CIFAR100. The results are as follows:
>
> |           Settings              |MW-Net   |GDW   |
> |:----:|:----:|:----:|
> |CIFAR10 + unif $0.4$ + imb $0.1$| $71.54$    |$76.30$  |
> |CIFAR10 + unif $0.6$ + imb $0.1$| $61.62$    |$70.24$  |
> |CIFAR10 + unif $0.4$ + imb $0.01$| $48.04$    |$48.53$  |
> |CIFAR10 + unif $0.6$ + imb $0.01$| $39.51$    |$40.07$ |
> |CIFAR100 + unif $0.4$ + imb $0.1$ | $36.10$    |$38.20$  |
> |CIFAR100 + unif $0.6$ + imb $0.1$| $24.80$    |$25.40$  |
> |CIFAR100 + unif $0.4$ + imb $0.01$|$21.26$    |$22.07$  |
> |CIFAR100 + unif $0.6$ + imb $0.01$|$12.75$   |$14.15$  |
>
> GDW also shows great performance gain over MW-Net, which means GDW can simultaneously tackle both problems.
>
> ### 5. Minor Comments
>
> - About Co-teaching: Thanks for the suggestion. For the clarity of the tables in our paper, we abbreviate "Co-teaching" as "Co-teach". We will revise this in the final version.
>
> - About "class imbalance": Thanks for the suggestion. The terms used in our work mostly follow the previous work on data weighting. We will check carefully about the correctness of using this term in our work.
>
> ### 6. Limitations and Social Impact
>
> Thanks for the suggestion. We will add the two parts in the final version.

---

### Official Review · Reviewer_tepP · 2021-07-10

**Rating:** 5
**Confidence:** 4

**Summary:**

This paper proposes a new method for mitigating the problem of label noise and class imbalance. The key idea is to utilize the class-level information to fully exploit the information contained in each training instance. The proposed method is evaluated using the popular dataset and it showed improved performance over the existing methods.

**Limitations And Societal Impact:**

No.
- It is recommended to discuss the limitation of the method more deeply in the main body of the paper. For example, the proposed method is tested only in label-noise case and in class-imbalance case separately. I think this is one of the limitation of the paper.
- The authors did not discuss any potential negative societal impacts. Therefore, the answer to this question in the checklist should be [No] rather than [N/A]. Any kinds of works can be used in an undesirable manner regardless of the authors’ intention. For example, the method may be used to intentionally increase the bias since it can handle class-level information. It is important to discuss the possible negative aspects (and hopefully how to cope with them).



**Main Review:**

# Strength
I found the key idea of utilizing class-level information within each instance is novel and interesting. In addition, the paper is basically well written except the part I point out below.  The method is evaluated using popular benchmark datasets and shows comparable or better performance compared to the existing state-of-the-art methods.

# Weakness
My main concerns are two folds.

## 1. The supports for the claims are not sufficient.
### 1.1 The coexistence of label noise and class imbalance and not explicitly handled in the experiment.
The authors repeatedly claim that one of the important problems in the real-world setting is the coexistence of label noise and class imbalance (L1, L24-25), and their GDW simultaneously mitigates these problems (L6, L300-301). However, these two issues are handled separately in the experiment, which is not good for supporting the above-mentioned claim.
### 1.2 The effectiveness of the proposed method in class-imbalance case seems marginal
Looking at the standard deviations, the proposed method is not significantly better than the existing methods. Is there any analysis on why the method works fine in label-nose setting whereas shows relatively lower performance in class-imbalance setting?

### 1.3 Figure 5 and L229-232
The meaning of “target” is not clear as I later point out in more detail. Therefore, it is not clear for me what $w_{nt}^n$ includes. L232 says it represents the importance of the “not bird”, but how about the “dog”? I think decreasing the weight of the “dog” gradient is as important as decreasing the weight of “not cat” in the case of Figure 1.

In addition, it is better to show $w_{tt}^c$, too, for the completeness.
### 1.4 Figure 6
How about the ratio trend of the number of increased $w_8$ on C1-C7, and C8? I think C8 instances should receive lower weights not only in the shown case, but also when C1-C7 instances are used in the training. Similarly, they should receive larger weights when C8 instances are used in the training. How is the same analysis for $w_9$ on C1-C8 and C9? Showing only one example is not sufficient.

### 1.5 L248-249: “We have also tried training without the zero-mean constraint in Section 3.3 and got poor results.”
Just “got poor results” is not enough as an evidence of the importance of zero-mean constraint. Please show it quantitatively.

## 2. Some parts of the paper are not clear enough.
### 2.1 L131-132 `Specifically, GDW tries to downweight the gradient flows for "dog" and "not cat", and upweight the gradient flow for "not bird".’
I understand what the proposed method aims to do. However I do not think the reason why such a thing is possible is clearly explained. Please explain the intuition behind this.
### 2.2 Section 3.3 is not clear
- L136: What does “retain the Softmax-CrossEntropy loss structure” mean? What is the motivation to do so?
- L138: What does “the target (label) position” mean? Especially in noisy label case, does “target” mean a true category or wrongly labeled ground-truth category? For example in Figure 1, what does “target” represent? Without precisely understand what is “target”, I cannot judge the soundness of equation 3 and subsequent discussion. For now, I can only understand equation 3 holds true if $w_t = w_j$.
### 2.3 equation 11
I think it is better to explicitly write down what $g_i$ exactly denotes in the equation.



**Time Spent Reviewing:**

10 hours

---

> ### Author Response · Authors · 2021-08-09
> **Response to review question**
>
> ### 0. General Reply
>
> Many thanks for your valuable and constructive comments on clarifying, correcting, and improving the materials in this paper! We will carefully revise the paper according to your comments as explained below.
>
> ### 1. The Supports for the Claims
>
> #### 1.1 The coexistence of both problems
>
> Thanks for the comments. To show that GDW can address both problems, we adopt the control variable method on the noise ratio and the imbalance factor. In this way, we show the effectiveness of GDW on each problem.
>
> Furthermore, we did comparison experiments under the co-existing setting after receiving this comment. Specifically, we compared with the mostly second-best method Meta-Weight-Net [23] under the mixed setting of uniform noise and class imbalance on CIFAR10 and CIFAR100. The results are as follows:
>
> |           Settings              |MW-Net   |GDW   |
> |:----:|:----:|:----:|
> |CIFAR10 + unif $0.4$ + imb $0.1$| $71.54$    |$76.30$  |
> |CIFAR10 + unif $0.6$ + imb $0.1$| $61.62$    |$70.24$  |
> |CIFAR10 + unif $0.4$ + imb $0.01$| $48.04$    |$48.53$  |
> |CIFAR10 + unif $0.6$ + imb $0.01$| $39.51$    |$40.07$ |
> |CIFAR100 + unif $0.4$ + imb $0.1$ | $36.10$    |$38.20$  |
> |CIFAR100 + unif $0.6$ + imb $0.1$| $24.80$    |$25.40$  |
> |CIFAR100 + unif $0.4$ + imb $0.01$|$21.26$    |$22.07$  |
> |CIFAR100 + unif $0.6$ + imb $0.01$|$12.75$   |$14.15$  |
>
> GDW also shows great performance gain over MW-Net, which means GDW can simultaneously tackle both problems.
>
> #### 1.2 The effectiveness in class-imbalance scenario
>
> Thanks for the comments. To further evaluate the performance of GDW, we conduct new experiments in the mixed setting, as shown in 1.1. The experiments in the mixed setting can prove the effectiveness of GDW.
>
> #### 1.3 Figure 5 and L229-232
>
> Thanks for the good question. "Target weights" means the class-level weights on the label position. "True-target weights" means the class-level weights on the true label position, which is only applicable for noisy instances. "Non-target weights" means the class-level weights on other positions.
>
> Therefore, $\omega_{nt}^n$ contains the class-level weights on the positions except the label position and the true label position.
>
> The importance of "dog", which is $\omega_t$ in our paper, is shown in **Figure 3**. The $\omega_t$ of clean instances are generally larger than those of noisy instances.
>
> According to the definition above, $\omega_{tt}^c$ is equivalent to $\omega_t^c$, which is already discussed in **Figure 3** and L224-227.
>
> We will add a more specific definition of these terms in the final version.
>
> #### 1.4 Figure 6
>
> Thanks for your suggestion. Due to the page limit, we only select the most representative weight trend, i.e. $\omega_8$ on C9, to show the difference of class-level weights and instance weights. We will put other trend curves in the appendix of the final version.
>
> #### 1.5 L248-249
>
> Thanks for the suggestion. When training without the zero-mean constraint, the accuracy converges to random prediction after several epochs. This is why we did not put the results in the paper. We will quantitatively show this in the final version of this paper.
>
> ### 2. Clarity
>
> #### 2.1 L131-132
>
> Thanks for the good question. We claim that GDW is possible to down-weight harmful gradient flows and up-weight useful ones. This is because GDW is guided by a clean and unbiased meta-set. By leveraging the golden information from the meta-set, GDW is possible to differentiate clean and noisy instances and thus control the importance of different gradient flows by the class-level weights.
>
> #### 2.2 Section 3.3
>
> - L136: Thanks for the good question. The structure of the gradient of *Softmax+CrossEntropy* is $p-y$ where $p$ is the probability and $y$ is the one-hot label. "Retain the structure" means to set the second term in Equation (13) to zero. We analyze the impact of not retaining the structure in Line 248-256.
> - L138: The meaning of "target" is the same as described above in 1.3. We will add the definition of these terms in the final version.
>
> #### 2.3 Equation 11:
> Thanks for the suggestion. The definition of $g_i$ is simply $$
>  \frac{1}{m} \sum_{i=1}^m \nabla_{\omega_i} l_{meta}(x_i^v, y_i^v; \boldsymbol \theta_\tau(\boldsymbol \phi_{\tau-1}))
> $$
> which is the intermediate gradient during the back-propagation on meta-set. We will explicitly write it in the final version.
>
> ### 3. Limitations and Social Impact
>
> Thanks for reminding us of the importance of writing limitations and possible negative social impacts. We will add the two parts in the final version.

---

> > ### Comment · Reviewer_tepP · 2021-08-13
> > **Thanks for the feedback**
> >
> > I appreciate the authors' feedback.
> >
> > ### 1.1
> > Thanks for the additional results. The results addressed my concern on this point.
> >
> > ### 1.2
> > Thanks for the feedback, but unfortunately it did not answer my question directly.
> >
> > ### 1.3
> > > "Target weights" means the class-level weights on the label position.
> >
> > Unfortunately this answer does not clarify yet what the "Target weights" is for me.
> > What is "the label position"?
> > > Therefore, $w^n_{nt}$ contains the class-level weights on the positions except the label position and the true label position.
> >
> > Again what does "the label position" mean?
> > This sentence implies that "target" includes two positions because "non target" excludes two positions, which confuses me even more.
> >
> > I think the authors should very carefully and clearly define the word "target".
> > If it has different meaning in the case of clean labels and noisy labels, it must be explicitly explained (and possibly it is better to represent them using different notations).
> >
> > > According to the definition above, $w^c_{tt}$ is equivalent to $w^c_t$, which is already discussed in Figure 3 and L224-227.
> >
> > I understand $w^c_{tt}$ is discussed in Figure 3, but my suggestion is it is better to show the trend of $w^c_{tt}$ in Figure 5 for the comparison with others.
> >
> > ### 1.4
> > Unfortunately the answer did not address my concern as it does not provide new information for me to judge if the claim in Figure 6 is well-supported or not.
> >
> > ### 1.5
> > > the accuracy converges to random prediction after several epochs.
> >
> > I think this is one of the important results.
> > I recommend providing analysis why this phenomenon happens in addition to the qualitative results.
> >
> > ### 2.1
> > > This is because GDW is guided by a clean and unbiased meta-set.
> >
> > This explanation did not clean up my doubt.
> > L154-155 and Figure 2 tell that the input of the three-layer MLP is (sum of) the loss of an training instance, which is a single scaler value.
> > The intuition on why the MLP is expected to be able to learn class-wise importance from the single value should be more clearly explained.
> > I think this question is related to the comment by reviewer B8Dp
> > > Furthermore, it looks quite non-trivial how to exploit class-level weights in front of noisy labels and imbalanced classes.
> >
> > In addition to the qualitative explanation on the above point, it is good to have more convincing experimental evidences. For example, something like Figure 1 (d)-(f) of MW-Net paper is a reasonable evidence to show how MW-Net reacts to different datasets. Figure 3-6 of the present paper partly provide the evidence, but not sufficient due to their incompleteness (1.3 and 1.4 in my comments. Furthermore, there are only results of a noisy-label case).
> >
> > ### 2.2
> > L136: Thanks for the clarification. I understand what the authors wanted to mean. Since what is meant here by "structure" and "retain" is something unusual (it seems reviewer B8Dp also does not understand them), the authors need to provide clear explanation in the manuscript.
> >
> > L138: The meaning of the "target" is still unclear for me as I mentioned above.
> > I still cannot judge the soundness of equation 3 and subsequent discussion.
> > For now, equation 3 holds true only when j=t.
> >
> > ### 2.3
> > Thanks for the clarification.

---

> > > ### Author Response · Authors · 2021-08-14
> > > **Response to review question**
> > >
> > > ### General Reply
> > >
> > > Many thanks for your valuable and constructive comments on clarifying, correcting, and improving the materials in this paper! We will carefully revise the paper according to your comments as explained below.
> > >
> > > ### 1.2
> > > > Is there any analysis on why the method works fine in label-nose setting whereas shows relatively lower performance in class-imbalance setting?
> > >
> > > Thanks for the comments. From the information perspective, GDW can better utilize class-level information under class imbalance, as supported by Figure 6 and the table in 1.4. However, we suppose that better information utilization may not result in large performance gain. Some factors, like the dataset size, the number of classes and the backbone network, can limit the improvement of performance under class imbalance. Therefore, we are now running experiments on datasets with larger number of classes. If time permits, we will report the comparison results in the final version of this paper.
> > >
> > >
> > > ### 1.3
> > > > Unfortunately this answer does not clarify yet what the "Target weights" is for me. What is "the label position"?
> > >
> > > Thanks for the comments. Since the class-level weight is a vector in $\mathcal{R}^C$ for each instance, the "weight at $i$'s position" simply means the $i^{th}$ element in that vector multiplied with the gradient $\frac{\partial \mathcal{L}}{\partial l_i}\frac{\partial \boldsymbol l_i}{\partial \boldsymbol \theta}$. For example, as shown in Figure 1 where a cat is mislabeled as "dog", the corresponding meanings of the notations are as follows:
> > >
> > >  - $\omega^n_{t}$ means $\omega_{dog}$ ("dog" is the target);
> > >  - $\omega^n_{tt}$ means $\omega_{cat}$ ("cat" is the ture target);
> > >  - $\omega^n_{nt}$ means $\omega_{bird}$ ("bird" is one of the non-target);
> > >
> > >
> > > For a correctly labeled cat, the corresponding meanings are:
> > >
> > > - $\omega^c_{t}=\omega^c_{tt}$ means $\omega_{cat}$; ("cat" is both the target and the ture target)
> > > - $\omega^c_{nt}$ means $\omega_{dog}$ and $\omega_{bird}$; ("dog" and "bird" are both non-target)
> > >
> > > > I think the authors should very carefully and clearly define the word "target". If it has different meaning in the case of clean labels and noisy labels, it must be explicitly explained (and possibly it is better to represent them using different notations).
> > >
> > > Thanks for the suggestion. "target" has the same meaning in both cases, while "non-target" is different. To be specific, non-target weights on a clean instance means all class-level weights excluding the target weight, while non-target weights on a noisy instance means all class-level weights excluding the target weight and the true target weight. We will illustrate these with a formal definition and specific examples in the final version.
> > >
> > > > I understand $\omega_{tt}^c$ is discussed in Figure 3, but my suggestion is it is better to show the trend of $\omega_{tt}^c$ in Figure 5 for the comparison with others.
> > >
> > > Thanks for the suggestion. We will add this curve in the final version.
> > > ### 1.4
> > > > Unfortunately the answer did not address my concern as it does not provide new information for me to judge if the claim in Figure 6 is well-supported or not.
> > >
> > > | weight/class | $C_0$ | $C_1$ |$C_2$ | $C_3$ |$C_4$ | $C_5$ |$C_6$ | $C_7$ |$C_8$ | $C_9$ |
> > > | :-----:| :----: | :----: |:-----:| :----: | :----: |:-----:| :----: | :----: |:----: | :----: |
> > > | $\omega_0$ | 0.036 | 0.968 | 0.973 | 0.972 | 0.965 | 0.974 |0.972  |0.976  |0.956  | 0.973 |
> > > | $\omega_1$ | 0.887 | 0.095 | 0.912 | 0.929 | 0.907 |0.927  |0.911  | 0.922 | 0.910 | 0.920 |
> > > | $\omega_2$ | 0.848 | 0.844 | 0.141 | 0.839 | 0.822 |0.845  |0.818  | 0.847 |0.829  | 0.802 |
> > > | $\omega_3$ | 0.585 | 0.608 | 0.552 | 0.405 | 0.569 |0.541  |0.561  |0.559  |0.617  |0.596  |
> > > | $\omega_4$ |0.474  | 0.521 | 0.420 | 0.460 |0.509  |0.455  |0.456  |0.482  |0.467  |0.512  |
> > > | $\omega_5$ |0.291  | 0.261 | 0.288 | 0.252 | 0.309 | 0.701 | 0.303 | 0.267 |0.297  |0.257  |
> > > | $\omega_6$ |0.199  |0.189  |0.169  |0.198  |0.196  |0.222  |0.778  | 0.195 |0.207  |0.182  |
> > > | $\omega_7$ |0.117  |0.117  |0.105  | 0.084 | 0.115 | 0.079 | 0.126 | 0.920 |0.133  |0.090  |
> > > | $\omega_8$ |0.115  |0.124  |0.178  |0.185  |0.184  |0.174  |0.191  |0.181  |0.862  |0.137  |
> > > | $\omega_9$ |0.043  |0.050  | 0.064 | 0.061 |  0.074| 0.062 |0.097  |0.069  |0.040  |0.935  |
> > >
> > > We report the ratio of the number of increased $\omega_i$ on $C_j$ instances in one epoch. $C_0$ is the largest class and $C_9$ is the smallest class.
> > >
> > > Note that $\omega_i$ on $C_i$ contains the "is $C_i$" information in the dataset. As a result, $\omega_i$ on $C_i$ should be large for small classes and small for large classes. As shown above, $\omega_i$ on $C_i$ (the diagonal elements) increases from $0.036$ to $0.935$ as $i$ increases from $0$ to $9$.
> > >
> > > Note that $\omega_i$ on $C_j$ ($j \neq i$) contains the "not $C_i$" information in the dataset. If $i$ is a large class, then $\omega_i$ on $C_j$ ($j \neq i$) should be large and vice versa. For $\omega_i$ ($i=0, 1, 2, 3, 4$), $\omega_i$ on $C_j$ ($j \neq i$) are generally larger than $0.5$; For $\omega_i$ ($i=5, 6, 7, 8, 9$), $\omega_i$ on $C_j$ ($j \neq i$) are generally less than $0.5$;
> > >
> > > ### 1.5
> > > >  I recommend providing analysis why this phenomenon happens in addition to the qualitative results.
> > >
> > > Thanks for the suggestion. We analyze why removing the zero-mean constraint may hurt the model's performance in Line 248-256. We will put the quantitative results together with the analysis in the final version.
> > >
> > > ### 2.1
> > > > The intuition on why the MLP is expected to be able to learn class-wise importance from the single value should be more clearly explained.
> > >
> > > Thanks for the good question. The three-layer MLP alone cannot learn class-wise importance, and this is why it is only used in the first stage of the proposed weight generation scheme. After the MLP outputs a scalar weight, GDW calculates the gradient for each class-level weight according to the feedback from the clean and balanced meta-set. Based on both the scalar weight and the gradients, GDW calculates the final class-level weights, as shown in Eq. (11).
> > >
> > > > Figure 3-6 of the present paper partly provide the evidence, but not sufficient due to their incompleteness.
> > >
> > > Thanks for the comments. We report a more detailed analysis on the class-level weights in 1.4, and we will put it in the final version of the paper.
> > >
> > >
> > > ### 2.2
> > > > L136: Since what is meant here by "structure" and "retain" is something unusual (it seems reviewer B8Dp also does not understand them), the authors need to provide clear explanation in the manuscript.
> > >
> > > Thanks for the suggestion. We will provide a clear explanation in the final version.
> > >
> > > > L138: The meaning of the "target" is still unclear for me as I mentioned above. I still cannot judge the soundness of equation 3 and subsequent discussion. For now, equation 3 holds true only when j=t.
> > >
> > > Thanks for the comments. We will illustrate the meaning of "target" in 1.3 with a formal definition and specific examples in the final version. Moreover, we have a detailed mathematical deduction of Eq. (3) in Appendix A, which can help understand the zero-mean constraint.

---

> > > > ### Comment · Reviewer_tepP · 2021-08-17
> > > > **Thanks for the additional results**
> > > >
> > > > ## 1.2
> > > > Thanks for the clarification. I think this aspect should be fairly discussed in the main paper to properly show the limitation of the present method. The fact that the proposed method achieved (marginal) improvement in class-imbalance settings is good, but I believe in-depth analysis including the weakness of the present method is much more important.
> > > >
> > > > ## 1.3
> > > > Thanks for the clarification. I understood what is meant with the concrete examples.
> > > >
> > > > ## 1.4
> > > > Thanks for the additional efforts. I think the table looks more convincing than Figure 6. It is very helpful to understand the behavior of the proposed method in more detail.
> > > >
> > > > ## 2.2
> > > > Even after looking at the appendix, I think equation 3 holds true only when j=t.
> > > > From equation (2) to (3) in the appendix, isn't it assumed that j=t?
> > > > What happens if j is not the "target" class?

---

> > > > > ### Author Response · Authors · 2021-08-17
> > > > > **Response to review question**
> > > > >
> > > > > ### General Reply
> > > > >
> > > > > Many thanks for your valuable and constructive comments on clarifying, correcting, and improving the materials in this paper! We will carefully revise the paper according to your comments as explained below.
> > > > >
> > > > > ### 1.2
> > > > >
> > > > > > The fact that the proposed method achieved (marginal) improvement in class-imbalance settings is good, but I believe in-depth analysis including the weakness of the present method is much more important.
> > > > >
> > > > > Thanks for the suggestion. We will analyze this weakness and write it in the limitation part in the final version.
> > > > >
> > > > > ### 1.4
> > > > >
> > > > > > I think the table looks more convincing than Figure 6. It is very helpful to understand the behavior of the proposed method in more detail.
> > > > >
> > > > > Thanks for the suggestion. We will put the table in the experimental part in the final version.
> > > > >
> > > > > ### 2.2
> > > > >
> > > > > > Even after looking at the appendix, I think equation 3 holds true only when j=t. From equation (2) to (3) in the appendix, isn't it assumed that j=t? What happens if j is not the "target" class?
> > > > >
> > > > > Thanks for the good question. From equation (2) to (3) in Appendix A, if $j$ is not the "target" class, $\boldsymbol y_j=0$. Therefore, the second term in equation (2) will equal $0$. This is why it can be rewritten as $\omega_t \boldsymbol y_j$ in equation (3).
> > > > >
> > > > > We realize that this is non-trivial. We will give a more explicit explanation of this deduction in the final version.

---

> > > > > > ### Comment · Reviewer_tepP · 2021-08-18
> > > > > > **Thanks**
> > > > > >
> > > > > > Thank you for the clarification.
> > > > > > Majority of my concerns are addressed through the rebuttal.
> > > > > > As a result, I indeed became more positive about the paper.
> > > > > >
> > > > > > However, it unfortunately did not change my overall rating since there are so many things to be added and modified, and it is difficult to judge if the final quality of the paper is sufficient or not.
> > > > > > In my opinion the paper needs major revision rather than minor one.
> > > > > >
> > > > > > Despite of my rating, I found the core idea of the paper interesting.
> > > > > > I believe the paper becomes much better if the descriptions including mathematical derivation become clearer and claims become well-organized and supported by convincing experimental results.
> > > > > > I would like to strongly encourage the authors to polish the paper again and re-submit it to other venues including NeurIPS 2022.

---

### Official Review · Reviewer_B8Dp · 2021-07-12

**Rating:** 5
**Confidence:** 4

**Summary:**

The paper focuses on a technique to learning per-instance weights in training classification networks. The paper improves over the idea of Meta-Weight-Net [23] which adopts meta-learning to learn per-instance weights. According to the paper, training data labels can be noisy and class labels usually follow long-tailed distributions, therefore learning per-instance weights is expected to better exploit training data for learning a better classification network. The paper's improvement is based on an intuition that class-level weights have not but should have been used for better learning per-instance weights. The paper proposes Generalized Data Weighting (GDW) and claims to *simultaneously* mitigate label noise and class imbalance. However, it is not clear why the proposed method is "generalized", and experiments do not validate the method can *simultaneously* mitigate label noise and class imbalance, because experiments study the two aspects separately.


**Ethical Concerns:**

It is not clear whether the datasets used in this paper have ethical concerns. Especially the Clothing1M dataset has person faces and body, it seems a concern in using the dataset. The authors failed in stating ethical concerns.


**Ethics Review Area:**

["Inadequate Data and Algorithm Evaluation", "Privacy and Security (e.g., consent)", "Legal Compliance (e.g., GDPR, copyright, terms of use)"]

**Limitations And Societal Impact:**

The paper does not discuss any limitations or societal impacts in the paper. The authors should be aware of the limitations of their methods. Authors should clarify.

**Main Review:**

pros:

+ The paper positions itself as using data reweighting to simultaneously address class-imbalanced learning and learning with noisy labels. This problem is interesting (although authors failed in experimentally validating the method on this problem).

+ Learning per-instance weights is a reasonable approach to address either imbalanced learning or learning with noisy labels.


Below are some summarized cons, followed by detailed comments.

- Due to such a position (as simultaneously addressing class-imbalanced learning and learning with noisy labels), the paper failed to justify the method in this experiments. In contrast, the paper studies the two problems separately.

- While class-imbalanced learning can be improved by learning per-instance weights, the paper did not fully survey the literature of class imbalanced learning. For example, [ref1] is missing in the paper that shows a rather simple method greatly improves class-imbalanced learning. Given this, it is natural to question why the paper misses an important baseline -- using [ref1] and data reweighting to simultaneously solve imbalanced learning with noisy labels.  [ref1] Decoupling Representation and Classifier for Long-Tailed Recognition, ICLR 2020


- The paper is incremental in terms of techniques. It improves over Meta-Weight-Net [23] by incorporating class-level weights learning. Perhaps because of this, the method is called *generalized* data reweighting; however, the method is incremental rather than *generalized*.


- There are many parts of the paper hard to follow, as detailed below.

First, as the paper positions itself as "simultaneously mitigate label noise and class imbalance" (Line6), the paper fails to justify whether the proposed method can do it because experiments study the two subproblems separately. Furthermore, it looks quite non-trivial how to exploit class-level weights in front of noisy labels and imbalanced classes. For example, it is not clear how to weight data with noisy labels from rare classes, should the model give a large weight on them because they are from rare classes, or a small weight because they have noisy labels? Authors should delve into this aspect to better support the position of the paper.

Line35: The sentence is confusing "Due to the MLP, MWNet has better scalability on large datasets compared with INSW". It is not clear what INSW does, and why MLP improves MWNet. Authors should improve the writing.

Line37: The sentence is confusing -- "they can not fully utilize class-level information within each instance". What do authors mean by "fully utilize class-level information"? Why "they can not"? What did "they" do such that "they can not"? Authors should clarify. This makes the paper hard to follow.

Line39: It is confusing -- what is the classifier for computing gradients? Does this classifier compute gradients from "not X" in terms of class-X?

Line42: What are the "three kinds of information"? The authors did not define the "three kinds of information". It is hard to understand the paper.

Line43: It is confusing -- "downweighting the "not bird" gradient flow is a waste of information." What do authors mean by "a waste of information"? Can authors clarify?

Figure 1: It is confusing -- "Although the gradient flows for "dog" and "not cat" contain harmful information..." What do authors mean by "harmful information"? Can authors clarify?

Line66: It is unclear what the authors mean by "To the best of our knowledge, we are the first to propose class-level weighting on gradient flows." Because it is not clear what the difference is between gradient reweighting and loss reweighting. Can authors clarify?

Section 2.2: There is one missed citation that is important [ref1]. Authors should carefully survey the literature.
[ref1] Decoupling Representation and Classifier for Long-Tailed Recognition, ICLR 2020

Line87: Although tt is evocative to say "meta-learning methods view instance weights as hyper-parameters and dynamically update them via a meta set to avoid hyper-parameter tuning." Did the authors consider the hyperparameters in the meta-learning? What are the learning rates, momentum, schedules of learning rates and initialization in meta learning? In the paper, authors did not provide such details.

Line94: What is "dynamic regularization"? Can authors define it?

Line127: It is not straightforward to understand how D1 is derived. Necessary derivations are needed.

Line136: What do authors mean by "retain Softmax-CrossEntropy loss structure after the manipulation"? What is structure? What will happen if one does not "retain" the structure? Authors should improve writing the make the paper readable.

Line140: What do the authors mean by "messes the structure"? What structure and how it "messes the structure"?

Eq.(4): The authors proposes a zero-mean constraint (Line143), but from what the authors did, it seems that this is factually not a constraint but an assumption with which the authors simply drop the second term. Authors should be careful in writing the statements.

Line213: The authors explain "GLC estimates the label corruption matrix well and thus performs the best", but why GLC does not estimate label corruption matrix well in other noise levels? Authors should provide insights or analysis.

Line229: As authors did not define "non-target weights", "true-target weights" or "non-target", I cannot understand this part. Authors should improve the writing quality.

Line261: The definition of the imbalanced factor is inconsistent. In plain words, it is defined as "the number of instances in the largest class divided by that of the smallest", that said, the factor should be no less than 1. However, authors states the factor is in (0,1). Authors should correct this inconsistency.







----------- post rebuttal --------------

I appreciate the authors' feedback! The answers largely addressed most my confusions, though I still have some below.

First, I think the explanation in the rebuttal is much clearer than the paper! Therefore, I'd upgrade the rating to "5: Marginally below the acceptance threshold". While this is not a recommendation for accepting the current version of the paper, I encourage the authors to incorporate the new results to polish the paper for the next submission.

Importantly, it is still confusing to connect the position of the paper and the proposed methods. Concretely, the position of the paper is "to simultaneously mitigate label noise and class imbalance" (line6 in the abstract) because "Label noise and class imbalance are two major issues coexisting in real-world datasets" (line1 in the abstract). This is a very good problem setup, but the authors only study this setup in the rebuttal.

Because this is a novel problem setup, thoughtful baselines should be compared, such as combining noisy training atop of a state-of-the-art (SOTA) class-imbalanced learning method [ref1] and combining class-imbalanced learning atop of a SOTA noisy training method. I'm not satisfied by the authors argument "it is not our duty to consider it as a baseline for comparison" -- this will hurt the paper. I have some background of long-tailed recognition, as a reader, I would think such a combination as the first method to try out to solve this novel problem, rather than rushing to any complicated methods.

On the other hand, in the rebuttal, the authors make another argument by "Should GDW perform better than this combination, the soundness of the comparison itself would be suspected", implying that the authors are reluctant to compare these straightforward baselines to address the novel problem setup. If the authors really want to sell the method, then authors can change the position of the paper -- instead of proposing a novel problem setup, authors can directly start by methods without emphasizing the problem.

Lastly, I do encourage the authors to make a thoughtful revision of the paper by considering the new experiment results, position of the paper, and comparisons to other baselines.

**Needs Ethics Review:**

Yes

**Time Spent Reviewing:**

4 hours

---

> ### Author Response · Authors · 2021-08-09
> **Response to review questions**
>
> ### 0. General Reply
>
> Many thanks for your valuable and constructive comments on clarifying, correcting, and improving the materials in this paper! We will carefully revise the paper according to your comments as explained below.
>
> ### 1. Simultaneously Address Both Problems
>
> Thanks for the comments. To show that GDW can address both problems, we adopt the control variable method on the noise ratio and the imbalance factor. In this way, we show the effectiveness of GDW on each problem.
>
> Futhermore, we did comparison experiments under the co-existing setting after receiving this comment. Specifically, we compared with the mostly second-best method Meta-Weight-Net [23] under the mixed setting of uniform noise and class imbalance on CIFAR10 and CIFAR100. The results are as follows:
>
> |           Settings              |MW-Net   |GDW   |
> |:----:|:----:|:----:|
> |CIFAR10 + unif $0.4$ + imb $0.1$| $71.54$    |$76.30$  |
> |CIFAR10 + unif $0.6$ + imb $0.1$| $61.62$    |$70.24$  |
> |CIFAR10 + unif $0.4$ + imb $0.01$| $48.04$    |$48.53$  |
> |CIFAR10 + unif $0.6$ + imb $0.01$| $39.51$    |$40.07$ |
> |CIFAR100 + unif $0.4$ + imb $0.1$ | $36.10$    |$38.20$  |
> |CIFAR100 + unif $0.6$ + imb $0.1$| $24.80$    |$25.40$  |
> |CIFAR100 + unif $0.4$ + imb $0.01$|$21.26$    |$22.07$  |
> |CIFAR100 + unif $0.6$ + imb $0.01$|$12.75$   |$14.15$  |
>
> GDW also shows great performance gain over MW-Net, which means GDW can simultaneously tackle both problems.
>
>  Additionally, the reason why GDW can *simultaneously* address both problems lies in the design of class-level weights, as explained below:
>
> Denote $\mathcal{L}$ as the loss and $l_i$ as the $i$-th logit. The gradient $\frac{\partial\mathcal{L}}{\partial l_i}\frac{\partial l_i} {\partial \theta}$ represents the information "This instance does (not) belong to class $i$". For convenience, we denote the information as "(not) $i$". This information has different interpretations in the two aspects.
>
> 1. Label noise: for a noisy instance labeled as $i$ whose true label is $t$, it contains information "$i$" and "not $j$" for all $j\neq i$. Among them, to our expectation, "$i$" and "not $t$" should be down-weighted because this information is wrong. Other information "not $j$" for all $j\neq i,t$ is still correct information which should not be down-weighted.
> 2. Class imbalance: the goal for class-level weights is to balance two kinds of information: "$i$" and "not $i$" for each class $i$. "$i$" is contained only in class $i$ instances, while "not $i$" is contained in all classes $j\neq i$. To balance the two kinds of information, GDW leverage class-level weights to adjust the overall distribution of classes.
> 3. It can be observed from the above two points that weighting under label noise is determined by the instance itself,  while weighting under class imbalance is determined by the overall distribution of classes. In this way, data weighting is decomposed into two parts, and they independently contribute to the final weights.
> 4. According to point 3, we can now answer the question "How to weight data with noisy labels from rare classes?". The answer is "It depends." For such a noisy instance labeled as $i$ ($i$ is the smallest class) whose true label is $t$, if the impact of data noise is larger, GDW will possibly give small class-level weights to "$i$" and "not $t$". Otherwise, GDW may give medium or large weights. GDW is guided by a clean and unbiased meta-set, so the weighting results should also be comprehensive considering both aspects. We conduct experiments on CIFAR10 with a 40% unif noise and different imbalance ratios $1.0$, $0.2$, $0.1$ and $0.01$, and record the ratios of the number of increased weights of "not $t$" and  "$i$"  to the number of all corresponding weights within an epoch.
> | Imbalance factor | Increased ratio of "not $t$"(%)  | Increased ratio of "$i$"(%)|
> | :----:| :----: | :----: |
> | 1.0 | 42.10  | 45.57 |
> | 0.2 | 63.26 | 48.24 |
> | 0.1 | 86.15 | 57.04 |
> | 0.01 | 95.59 | 64.58 |
>
>       As shown above, given a balanced dataset (imbalance factor = $1.0$), less than 50% class-level weights of  "not $t$" and  "$i$" increase. As the imbalance factor decreases from $1.0$ to $0.01$, more class-level weights increase ("not $t$" from 42.10 to 95.59; "$i$" from 45.57 to 64.58).
>
> ### 2. Missed Citation
>
> Thanks for the comments. We will add this citation in the final version of our paper.
>
> It is an interesting idea to combine the classifier adjustment methods with data reweighting to tackle both problems. We will investigate this interesting combination and write another paper if it works. However, to the best of our knowledge, no existing literature suggests such a combination. Therefore, it is not our duty to consider it as a baseline for comparison. On the one hand, the performance of this combination is not solidly verified. Should GDW perform better than this combination, the soundness of the comparison itself would be suspected. On the other hand, investigating the combination of class imbalance methods and data reweighting methods is beyond the scope of our work. We only select existing state-or-the-arts for comparison.
>
> ### 3.1 About *Generalized*
>
> Thanks for the good question. We explained in Line 59 and 125 that Meta-Weight-Net [23] is a special case of GDW. [23] assign each instance with a scalar weight while we assign each instance with a class-level (vector) weight. Moreover, we apply the class-level weights directly to the gradients instead of the loss, which makes data weighting unaware of the form of loss functions. These are the reasons why we call out method *Generalized* Data Weighting.
>
> ### 3.2 Main Technical Points
>
> The main contribution of our work is that we propose class-level data weighting instead of instance-level data weighting. This is not just an increment work of [23]. We introduce a complete framework to integrate this novel idea into data weighting, and we conduct extensive experiments to verify our framework. Although the Bilevel Optimization algorithm in our work is similar to that in MW-Net [23], GDW is different from [23] in the following three aspects:
>
> 1. [23] propose to reweight each data instance by applying the weights to the loss. However, as explained in Line 36-46, instance weighting cannot fully utilize class-level information. Instead, we illustrate the connection between the gradient flow and the class-level information in **Figure 1** and propose to reweight class-level gradient flows directly. This intuition results from the correlation between class-level weights and class-level gradient flows illustrated in Section 3.2 through mathematical deduction.
> 2. It is non-trivial to make class-level reweighting on gradient flows work, because simply reweighting gradient flows leads to poor results. Therefore, in Section 3.3, we propose an important constraint on the class-level weights. This constraint is to regularize the weights and improve training, as illustrated in Line 248-256. However, [23] do not have any regularizations on the instance weights.
> 3. [23] uses a three-layer MLP to generate instance weights. Although we also adopt the same network, we propose a two-stage weight generation scheme to generate class-level weights, a vector for each instance instead of a scalar. The novelty of this weight generation scheme is that we utilize the intermediate gradients on the class-level gradients, which does not introduce any extra computational cost or trainable parameters, compared with [23].
>
> ### 4. Paper Writing
>
> Thanks for your suggestions on the paper writing. We will revise the paper writing according to your comments. We also receive positive comments about the paper writing and paper organization from other reviewers. Generally speaking, the overall writing quality of this paper is not bad.
>
> The remaining reply is on the next page due to the page limit.

---

> > ### Author Response · Authors · 2021-08-09
> > **The remaining response to review questions**
> >
> > ### 5. Minor Comments
> >
> > - Line 35: Thanks for the good question. INSW [24] assign each instance a weight while MW-Net [23] generates weights using the MLP network. Therefore, the number of trainable parameters is linearly increased with the dataset size in [24], while it is fixed in [23]. Therefore, we claim that [23] have better scalability. We will explain this point more clearly in Related Work.
> > - Line 37: Thanks for the good question. Because these instance weighting methods only assign each instance a scalar weight, and the drawback is that these methods fail to utilize some of the useful information in noisy instances, as explained in Line 38-46 by an explicit example. We will clarify the statement in the final version.
> > - Line 39: Thanks for the good question. The gradient is computed as $\frac{\partial\mathcal{L}}{\partial \mathbf{l}} \frac{\partial \mathbf{l}}{\partial \mathbf{\theta}}$ where $\mathcal{L}$ is the loss and $\mathbf{l}$ is the logits (output of the classifier). Therefore, the calculation does not require to specify the classifier.
> > - Line 42: Thanks for the good question. The three kinds of information are "not cat", "dog" and "not bird", which is explained in both Line 40-41 and **Figure 1**.
> > - Line 43: Thanks for the good question. Since the instance contains the useful information "This image is not a bird", the gradient flow corresponding to this information should not be down-weighted.
> > - **Figure 1**: Thanks for the good question. The two kinds of information "This image is a dog" and "This image is not a cat" are both wrong information for model training, which can degrade the classification performance. Therefore we call them "harmful information".
> > - Line 66: Thanks for the good question. We adopt the cross-entropy loss function, which is the default and effective loss function for most single-label classification tasks. In this cross-entropy loss function, there is no individual loss term for every class, so we can not reweight the loss per class directly. Furthermore, since most neural network training adopts SGD and its variants for optimization, applying weights to gradients directly can more clearly reflect the importance of each gradient flow, which corresponds to some interpretable class-level information, as shown in **Figure 1** and **Figure 4**. Moreover, applying weights to gradients is unaware of the loss functions, which means GDW can also be combined with different loss function designs for better performance.
> > - Section 2.2: Thanks for the comments. We will cite this important paper in the final version.
> > - Line 87: Thanks for the good question. See appendix B for the hyperparameter details of this work. We also explain the training details in the **Training** subsection of each setting.
> > - Line 94: Thanks for the good question. Soft label [16] is a good regularization for the classification model since it reduces the confidence of the model prediction. [37, 39] propose to learn dynamic soft labels for each instance. This is why we call these methods "dynamic regularization".
> > - Line 127: Thanks for the suggestion. This conclusion requires a derivative operation of the softmax function and the cross-entropy loss function. We will illustrate the deduction in the appendix.
> > - Line 136 & 140: Thanks for the good question. The structure of the gradient of *Softmax+CrossEntropy* is $p-y$ where $p$ is the probability and $y$ is the one-hot label. "Messes the structure" means the second term in Equation (13) is not zero. We analyze the impact of not retaining the structure in Line 248-256.
> > - Eq. (4): Thanks for the comments. This is indeed a constraint because we modify the class-level weights according to this constraint, as shown in the right part of Equation (4). The usage of this constraint is also illustrated in **Algorithm 1**. The statement here is correct.
> > - Line 213: Thanks for the good question. GLC [19] estimates a label corruption matrix to correct the prediction. This estimation is quite easy under the 40% flip noise (flip one class to another class w.p. 40%). This is why GLC gets better results. However under uniform noise scenarios, as shown in **Table 2**, GLC yields bad performance compared to GDW. This is because uniform noise is hard to estimate.
> > - Line 229: Thanks for the comments. "Target weights" means the class-level weights on the label position. "True-target weights" means the class-level weights on the true label position, which is only applicable for noise instances. "Non-target weights" means the class-level weights on other positions.
> > - Line 261: Thanks for the comments. This is a typo. The definition of imbalance factor should be "the number of instances in the smallest class divided by that of the largest", which is in (0, 1). We will revise this in the final version.
> >
> > ### 6. Limitations
> >
> > Thanks for the suggestion. We wrote it in the Checklist: "The proposed method can only be applied on classification tasks." We will add this part in the final version.
> >
> > ### 7. Ethical Concerns
> >
> > Thanks for the comments. We conduct experiments on Clothing1M because the Clothing1M is a well-known benchmark dataset for image classification with noisy labels. Another reason is to have a fair comparison with previous work such as meta-weight-net[23] which also conducted experiments on Clothing1M. We can not judge and confirm whether there exist ethical issues in Clothing1M, and we will remove the experiments related to Clothing1M in the final version if the editors and chairs find there are ethical problems in Clothing1M.

---

> > > ### Comment · Reviewer_B8Dp · 2021-08-14
> > > **post-rebuttal**
> > >
> > > I appreciate the authors' feedback! The answers largely addressed most my confusions, though I still have some below.
> > >
> > > First, I think the explanation in the rebuttal is much clearer than the paper! Therefore, I'd upgrade the rating to "5: Marginally below the acceptance threshold". While this is not a recommendation for accepting the current version of the paper, I encourage the authors to incorporate the new results to polish the paper for the next submission.
> > >
> > > Importantly, it is still confusing to connect the position of the paper and the proposed methods. Concretely, the position of the paper is "to simultaneously mitigate label noise and class imbalance" (line6 in the abstract) because "Label noise and class imbalance are two major issues coexisting in real-world datasets" (line1 in the abstract). This is a *very good* problem setup, but the authors only study this setup in the rebuttal.
> > >
> > > Because this is a *novel problem setup*, thoughtful baselines should be compared, such as combining noisy training atop of a state-of-the-art (SOTA) class-imbalanced learning method [ref1] and combining class-imbalanced learning atop of a SOTA noisy training method. I'm not satisfied by the authors argument "it is not our duty to consider it as a baseline for comparison" -- this will hurt the paper. I have some background of long-tailed recognition, as a reader, I would think such a combination as the first method to try out to solve this novel problem, rather than rushing to any complicated methods.
> > >
> > > On the other hand, in the rebuttal, the authors make another argument by "Should GDW perform better than this combination, the soundness of the comparison itself would be suspected", implying that the authors are reluctant to compare these straightforward baselines to address the novel problem setup. If the authors really want to *sell the method*, then authors can change the position of the paper -- instead of proposing a novel problem setup, authors can directly start by methods without emphasizing the problem.
> > >
> > > Lastly, I do encourage the authors to make a thoughtful revision of the paper by considering the new experiment results, position of the paper, and comparisons to other baselines.

---

> > > > ### Author Response · Authors · 2021-08-14
> > > > **Response to review questions**
> > > >
> > > > ### General Reply
> > > >
> > > > Many thanks for your valuable and constructive comments on clarifying, correcting, and improving the materials in this paper! We will carefully revise the paper according to your comments as explained below.
> > > >
> > > > ### 1
> > > > Thanks for your recognition!  We will incorporate the new results to polish the paper in the final version.
> > > >
> > > > ### 2
> > > > Firstly, to show that GDW can address both problems, we adopt the control variable method on the noise ratio and the imbalance factor. In this way, we show the effectiveness of GDW on each problem.
> > > >
> > > > Secondly, it is non-trivial to combine the SOTA method for label noise and the SOTA method for class imbalance. Besides, there are too many possible combinations. To the best of our knowledge, these combinations have never been considered by any previous work. Currently, we are working on all possible combinations and will write another research paper given good results.

---

### Official Review · Reviewer_iJG7 · 2021-07-19

**Rating:** 5
**Confidence:** 3

**Summary:**

The authors propose Generalized Data Weighting (GDW) to generalize data weighting from instance level to class level by reweighting gradient flows. This framework can be helpful for better information use when handling label noise and class imbalance.  Specifically, they design a two-stage weight generation scheme embedded in the three-step bi-level optimization framework which can efficiently obtain class-level weights.  This scheme leverages intermediate gradients to update class-level weights via a gradient descent step. Experiments show that GDW achieves remarkable performance improvement in various settings.

**Limitations And Societal Impact:**

Although the proposed algorithm seems to work well empirically, there is no theoretical analysis on the performance. Also, the main techniques this paper leverages have already been well-established in [23]. The technical contribution seems somewhat marginal.

**Main Review:**

Originality:
In contrast to the classical instance level weighting, the authors are the first to propose class-level weighting on gradient flows. However, the main techniques this paper leverages have already been well-established in [23] which proposed the MW-Net Learning Algorithm. This is a nice extension of the existing techniques.

Quality:
The authors conduct extensive experiments in various settings ( Label Noise Setting, Class Imbalance Setting, Real-world Setting) to verify the effectiveness of GDW.

Clarity:
This paper is organized well. I can easily understand the main idea and techniques applied

Significance:
This paper advances state-of-the-art methods to alleviate Label noise and class imbalance by leveraging class-level information. The algorithm proposed in this paper would be helpful to deal with two major issues coexisting in real-world datasets.


**Time Spent Reviewing:**

3 hours

---

> ### Author Response · Authors · 2021-08-09
> **Response to review questions**
>
> ### 0. General Reply
>
> Many thanks for your valuable and constructive comments on clarifying, correcting, and improving the materials in this paper! We will carefully revise the paper according to your comments as explained below.
>
> ### 1. Main Technical Points
>
> The main contribution of our work is that we propose class-level data weighting instead of instance-level data weighting. This is not just a simple extension of [23]. We introduce a complete framework to integrate this novel idea into data weighting, and we conduct extensive experiments to verify the effectiveness of our framework. Although the Bilevel Optimization algorithm in our work is similar to that in MW-Net [23], GDW is different from [23] in the following three aspects:
>
> 1. [23] propose to reweight each data instance by applying the weights to the loss. However, as explained in Line 36-46, instance weighting cannot fully utilize class-level information. Instead, we illustrate the connection between the gradient flow and the class-level information in **Figure 1** and propose to reweight class-level gradient flows directly. This intuition results from the correlation between class-level weights and class-level gradient flows illustrated in Section 3.2 through mathematical deduction.
> 2. It is non-trivial to make class-level reweighting on gradient flows work, because simply reweighting gradient flows leads to poor results. Therefore, in Section 3.3, we propose an important constraint on the class-level weights. This constraint is to regularize the weights and improve training, as illustrated in Line 248-256. However, [23] do not have any regularizations on the instance weights.
> 3. [23] uses a three-layer MLP to generate instance weights. Although we also adopt the same network, we propose a two-stage weight generation scheme to generate class-level weights, a vector for each instance instead of a scalar. The novelty of this weight generation scheme is that we utilize the intermediate gradients on the class-level gradients, which does not introduce any extra computational cost or trainable parameters, compared with [23].
>
> ### 2. Theoretical Analysis on Performance
>
> The theoretical analysis of the performance of bilevel optimization has long been a hard topic in research areas. The convergence conditions are illustrated in [31]. However, it is generally believed that neural networks do not satisfy those strict conditions.
>
> Therefore, instead of theoretical analysis, we provide sufficient experimental analysis on the performance of GDW. Specifically,
> 1. For the label noise setting,
>     1. We analyze the distribution of target weights in **Figure 3** to illustrate that GDW can differentiate noisy and clean data points.
>     2. We analyze the distribution of non-target weights and true-target weights in **Figure 5** to illustrate that GDW can properly adjust class-level gradient flows according to our intuition.
> 2. For the class imbalance setting, we analyze the distribution of class-level weights on the minor class to illustrate that GDW can adjust class-level weights according to the imbalanced distribution, which better handles class imbalance.

---

### Review · Ethics_Reviewer_ThFr · 2021-08-11

**Recommendation:** N/A - there are no issues to address.

**Ethics Review:**

The ethics review was initiated due to the following comment from Reviewer B8Dp.

``It is not clear whether the datasets used in this paper have ethical concerns. Especially the Clothing1M dataset has person faces and body, it seems a concern in using the dataset. The authors failed in stating ethical concerns.``

Having read through the paper and done background research on the dataset, this appears to be a false alarm.

The paper introduces a general-purpose technique and uses the Clothing 1M to demonstrate the functionality ​of this technique – rather than, e.g., building better facial recognition technology. The technique is not designed to be used in a way that would inflict harm when used as intended, misused, or abused (no more than any other deep learning technology).

Clothing1M is composed of images obtained by crawling online shopping websites, which raises some potential concerns regarding privacy and IP. I am not sure of the legal implications of collecting the dataset. That being said, I do not believe that the issue merits escalation or another review for the following reasons:

1. The images are likely to be public domain
2. The [CVPR paper](https://openaccess.thecvf.com/content_cvpr_2015/papers/Xiao_Learning_From_Massive_2015_CVPR_paper.pdf) has been cited 500+ times, which suggests that the dataset itself has been readily used by the computer vision community.

---

### Decision · Program_Chairs · 2021-09-28

**Decision:**

Accept (Poster)

**Comment:**

The paper studies the problem when data is imbalanced and has label noise. The authors proposed to simultaneously mitigate label noise and class imbalance. The key idea is to exploit the class-level weights, which is very interesting, reasonable, novel, and well-written. Experiments also show the effectiveness of the proposed method. No computational cost will be introduced. However, there are some concerns. (1) Insufficient experiments under the claimed setting to support the proposed method. The authors are expected to either add more experiments under the claimed setting (more than the experiments they presented in the rebuttal) or change the position of the paper writing. (2) The proposed method does not work that well in the cases with flip label noise. More discussions are needed to explain this. We also expect the authors to add experiments on synthesized instance-dependent label noise rather than uniform and pairflip label noise. (3) Thoughtful baselines such as combining some classic/state-of-the-art methods of training with noisy labels and class-imbalanced learning are expected. For example, the paper: Menon et al. Learning from Corrupted Binary Labels via Class-Probability Estimation, ICML 2015, which shows that the balanced error designed for imbalanced data is robust to label noise. This is a very related work which should be discussed as well. In a summary, we think the paper is not ready for publication at this round. It will be a strong paper if the concerns can be well addressed.

**Consistency Experiment:**

NeurIPS has a long history of experimentation. In 2014, NeurIPS ran an experiment in which 10% of submissions were reviewed by two independent committees to quantify the randomness in the review process. This year, we repeated a variant of this experiment to see how the quality of the review process has changed over time.  This paper was part of the experiment and was therefore assigned to two committees (consisting of reviewers, an Area Chair, and a Senior Area Chair) that reached independent decisions.  If both committees made the same recommendation, this recommendation was followed. If a single committee recommended acceptance, the paper was accepted (with the exception of a few cases in which the other committee identified what we considered a fatal flaw, e.g., an error in a key result).

This copy’s committee reached the following decision: **Reject**

The other committee assigned to the paper recommended **Accept (Poster)**.  You can find the other set of reviews, along with any follow up discussion with the authors here:
https://openreview.net/forum?id=xRrdX_wV1JI